# INVARIANCE THROUGH INFERENCE

## ABSTRACT

We introduce a general approach, called Invariance through Inference, for improving the test-time performance of an agent in deployment environments with unknown perceptual variations. Instead of producing invariant visual features through interpolation, invariance through inference turns adaptation at deployment-time into an unsupervised learning problem. This is achieved in practice by deploying a straightforward algorithm that tries to match the distribution of latent features to the agent's prior experience, without relying on paired data. Although simple, we show that this idea leads to surprising improvements on a variety of adaptation scenarios without access to deployment-time rewards, including changes in camera poses and lighting conditions. Results are presented on challenging distractor control suite, a robotics environment with image-based observations.

## 1 INTRODUCTION

Let us consider the ability of an intelligent agent to generalize to unseen domains. To have such a discussion, we must first consider what generalization means. In much of the learning literature, we typically assume that an agent will accrue experiences of sufficient variation during training, and that these experiences will allow the agent to generalize to novel settings during deployment. Since the richness of the agent's experience is of paramount importance to the quality of its generalization, there exist a broad family of methods that expand the support distribution of the agent's training set. Domain transfer, domain randomization, and meta learning all fall into this category.

Expanding the support of the training distribution is most often accomplished via artificial data augmentation. In pixel-based control tasks, for example, image observations are cropped, shifted, rotated, and discolored to make learned policies more robust to shifts in the input observation at test time (Hansen & Wang, 2020; Yarats et al., 2021). In RL, this type of augmentation can cause value function estimation to become unstable (Laskin et al., 2020; Raileanu et al., 2020; Kumar et al., 2021), and so care must be taken to avoid destabilizing training.

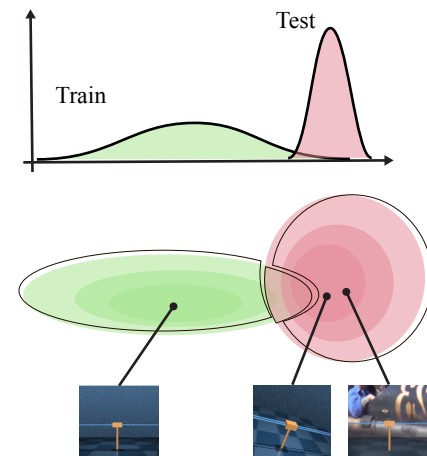

Figure 1: Methods such as data augmentation try to make the training distribution large at the expense of complexity and performance. In spite of these efforts, these methods still often fail to adequately cover the target distribution.

While these approaches are powerful, they are often prone to failure in the case where the target domain is not known *a priori*. In this case, if the target domain falls out-of-distribution, the agent will lack the ability to self correct and will fail, even if great pains were taken to significantly widen the size of its training support. Indeed, as we show in Section 5, even the most robust data augmentation schemes fail when the target observation space differs substantially from that of training. We refer to these approaches *Invariance through Interpolation*, which aims to improve an agent's generalization by increasing the richness of its experiences, as in-distribution generalization.

A more challenging problem—but one that is also more realistic—is generalizing out-of-distribution. In this setting, we assume the specifics of the agent's deployment are not known in advance. Taking this even further, we assume that access to instrumented reward supervision is unavailable in the deployment environment. Here, the burden placed on the agent hoping to generalize is significant. Since we do not have access to reward, fine-tuning via reinforcement learning is impossible. Prior experience will help, but a truly out-of-distribution environment of which the agent has no prior knowledge will make the transfer problem non-trivial. While difficult, this setting of out-of-distribution generalization is nevertheless pervasive in fields such as robotics, wherein a robot trained from pixel observations in the factory will inevitably encounter new settings with different lighting conditions, environmental objects, physical dynamics, and sensor modalities during deployment.

To make progress on out-of-distribution generalization, we consider how we can best leverage the information we do have: the agent's prior experiences during training. Specifically, if we assume the original task the agent was optimized for during training remains well-defined in the target domain, then it is also safe to assume that the agent does have some understanding of the underlying Markov decision process in the target domain. Without further assumptions or loss of generality, we may recast the out-of-distribution generalization problem into an unsupervised adaptation between two MDPs that share underlying dynamics and reward structure, but with distinct observations.

This type of unsupervised adaptation in RL has not been explored widely yet to the best of our knowledge. One of the simplest approaches in this vein is to train a policy with a self-supervised auxiliary task such as inverse dynamics, and rely entirely on it during adaptation (Hansen et al., 2020). Yet attractive for its simplicity, it often performs relatively poorly, as discussed in Section 5. Another approach is to enforce Cycle-GAN (Zhu et al., 2017) style objective to learn cycle-consistent mappings between domains (Zhang et al., 2020). Although it is shown to work well in multiple scenarios, it requires access to states, which is not generally available in pixel-based control tasks we consider.

In this paper, we investigate ways to improve generalization under this challenging scenario. Rather than producing policy invariance by explicitly baking this property into training, we harness probabilistic inference to produce invariance by taking advantage of the latent structure the agent has already discovered in the environment. As we will see, this pays great dividends in producing robotic agents that can function effectively in deployment environments that exhibit different lighting, object texture, or camera poses. To distinguish our approach from prior works that attempt to produce generalization by baking invariances into the policy at training time, we refer to our method as "*Invarivance Through Inference*."

## 2   RELATED WORK

**Generalization in reinforcement learning** is a longstanding problem. Recent work has shown that accurately quantifying generalization in this setting poses a challenge (Lee et al., 2019; Cobbe et al., 2019; Zhang et al., 2018). This challenge is further amplified in visual RL from pixels, where measuring the difference between two problems is especially difficult. One promising class of techniques for generalization in image-based RL utilize image augmentation to greatly increase the size of the training distribution. These techniques have enjoyed success in vision research (Chen et al., 2020a; He et al., 2020). However, it was only recently shown that such data augmentation leads to instability in value function estimation in RL algorithms (Hansen & Wang, 2020; Hansen et al., 2021). This line of work showed how to stabilize data-augmented RL, which lead to a significant increase in the generalization capabilities of RL algorithms. The primary shortcoming of these methods is that it is largely impossible to effectively cover the entire potential test distribution by expanding the training distribution, and some test time adaptation is often needed. Several recent works (Hansen et al., 2020; Wang et al., 2020) consider this setting of adaptation at test time, and are further discussed in Section 3.

**Invariant Representation Learning** is a broad class of techniques for learning representations that are robust across changes in the agent's environment or reward. Often, the agent will use auxiliary information related to dynamics (Hafner et al., 2020) or reward prediction (Jaderberg et al., 2016) to learn a latent representation that is invariant across changing environmental conditions. This is closely related to the idea of **self-supervised learning** (Chen et al., 2020b; Grill et al., 2020).

When applied to visual space RL, it has been shown that self-supervised pre-training provides an efficient way to bootstrap useful invariant representations, even in the absence of a ground-truth reward (Schwarzer et al., 2020; Srinivas et al., 2020).

**Domain Shift** considers the problem of test-time shift in the underlying input data distribution (Ganin et al., 2016). One common class of techniques for addressing this problem is to learn representations that maximize domain confusion, thus requiring that the agent learn domain-invariant features. This idea has been applied successfully in imitation learning (Stadie et al., 2017). More recent advances have combined Cycle-GANs with imitation learning to produce features that are cycle consistent across domains (Smith et al., 2019). Another approach bottlenecks the information on the latent code of a VAE-based architecture to learn features that are more robust across domain shift in a variety of adversarial settings (Peng et al., 2018).

**Meta Learning** tries to solve generalization by aggregating learning across distributions of tasks (Finn et al., 2017; Duan et al., 2016; Rakelly et al., 2019). Often, meta learning pipelines focus heavily on fast inference at test time. This is most often accomplished through using some form of bi-level optimization (Rothfuss et al., 2018; Houthooft et al., 2018). Unfortunately, meta learning methods generally require training over entire task distributions. Worse still, this results in the need for a distribution of reward functions for each task during both training and test time. Defining even a single reward function is often quite difficult, let alone many closely related reward functions. Recent work has tried to alleviate this burden with unsupervised training (Hsu et al., 2018).

## 3 PROBLEM FORMULATION

Consider the standard infinite horizon Markov decision process (MDP) (Puterman, 2014) parameterized via the tuple $\mathcal{M} = \langle S, O, A, R, P, \gamma \rangle$, where $S$, $O$, and $A$ are the state, observation, and action spaces, respectively, $P : S \times A \mapsto S$ is the transition function, $R : S \times A \mapsto \mathbb{R}$ is the scalar reward, and $\gamma$ is the discount factor. Recall that the state space $S$ usually represents ground-truth information about the agent's environment, which we assume the agent can not directly access. Instead, the agent receives a stream of observations $o \in O$ that convey information about the environment. In this paper, we assume the environment is fully-observable. In other words, the true state can be perfectly inferred from the corresponding observation, given the right feature extractor. The goal of reinforcement learning is to solve for the policy $\pi : O \times A \mapsto [0, 1]$ that maximizes the expected discounted return $\mathcal{J} = \mathbb{E}\left[\sum_{\infty} \gamma^t R(s_t, a_t)\right]$ over infinite horizon, represented as the $Q^\pi(o, a)$ function. In this paper, we assume that the policy $\pi$ consists of an encoder $F : O \mapsto Z$, where $Z$ is a compact latent space, and a policy $\pi_z : Z \times A \mapsto [0, 1]$.

We consider a setting wherein there are two distinct domains: a source domain $\mathcal{M}_{\text{src}}$ and a target domain $\mathcal{M}_{\text{tgt}}$. Crucially, when the agent is placed in the previously unseen target domain, we assume that it can only access its observations and the actions it took, but not reward. Often, we are most interested in the case where shift between $\mathcal{M}_{\text{src}}$ and $\mathcal{M}_{\text{tgt}}$ is induced by differences in the observation spaces $O_{\text{src}}$ and $O_{\text{tgt}}$. That is, the state and reward dynamics between the source and target domains are quite similar, but the observation spaces between the two are significantly different. Practically speaking, this can happen quite easily in the presence of distractors, corrupted or malformed inputs, or shifts in sensor readings at test time. Consider a robot that is trained from image observations in a clean environment, and then expected to perform the same task in the wild, where changing lighting conditions and environmental conditions might lead to significantly different image observations. As a result of this distributional shift in the observations on which the policy is conditioned, deploying an agent trained in the source domain directly in the target domain (i.e., zero-shot transfer) results in poor performance.

Given an agent pretrained on a source domain, our goal is to adapt it to the target domain. As we do not have knowledge about the correspondence of observations in the two domains, this setting essentially requires us to make use of unpaired trajectory samples from each domain to drive adaptation. The proposed method, Invariance through Inference, aims to only adapt the encoder $F$ in a manner that minimizes distributional shift in $Z$ between the domains and, in turn, enables the policy $\pi_z$ to transfer to the unseen target domain, by utilizing only unpaired transitions. We achieve this by jointly optimizing two objectives: *distribution matching* and *dynamics consistency*.

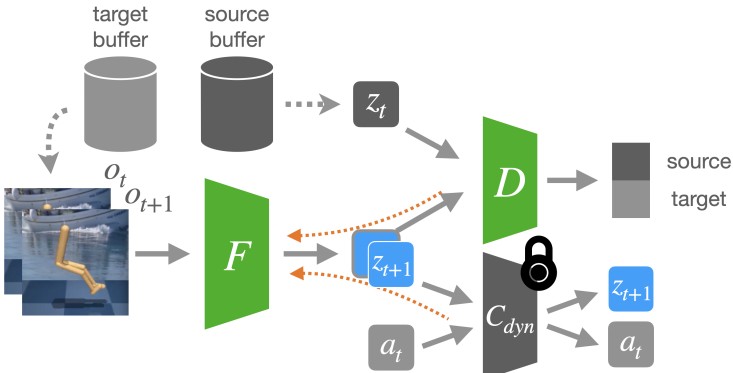

Figure 2: Our proposed Invariance through Inference architecture. The encoder $F$ takes an observation of target domain, and learns to fool the discriminator, while the discriminator $D$ predicts whether the input is an encoded target observation or a latent sample from source buffer. This adversarial training encourages the distribution of encoder outputs to be similar to the latent embedding sampled from the source buffer. $C_{\mathrm{dyn}}$ is the pretrained forward and inverse dynamics networks used only to guide the encoder during adaptation.

## 4 INVARIANCE THROUGH INFERENCE

### 4.1 DISTRIBUTION MATCHING

We first deploy a random exploration policy in both the source and target domains to collect samples of observation trajectories

$$\mathcal{B}_{\mathrm{src}} = \left(\ldots, o_t^{\mathrm{src}}, a_t^{\mathrm{src}}, o_{t+1}^{\mathrm{src}}, \ldots\right) \sim P_{\bar{\pi}}(\mathcal{M}_{\mathrm{src}}) \tag{1a}$$

$$\mathcal{B}_{\mathrm{tgt}} = \left(\ldots, o_t^{\mathrm{tgt}}, a_t^{\mathrm{tgt}}, o_{t+1}^{\mathrm{tgt}}, \ldots\right) \sim P_{\bar{\pi}}(\mathcal{M}_{\mathrm{tgt}}), \tag{1b}$$

where $P_{\bar{\pi}}(\mathcal{M})$ denotes the distribution of transitions produced with a random policy in domain $\mathcal{M}$. Given an observation $o_t$, an encoder $F$ produces a corresponding latent representation $z_t$

$$z_t^{\mathrm{src}} = F(o_t^{\mathrm{src}}) \tag{2a}$$

$$z_t^{\mathrm{tgt}} = F(o_t^{\mathrm{tgt}}). \tag{2b}$$

We seek to match the distribution over $z_t^{\mathrm{tgt}}$ with that over $z_t^{\mathrm{src}}$ by adapting the weights of the encoder $F$, without access to test-time rewards. We employ adversarial training to achieve this. Specifically, our architecture (Fig. 2) includes a discriminator $D$ that tries to distinguish between embeddings from the source domain $z_t^{\mathrm{src}}$ and those from the target domain $z_t^{\mathrm{tgt}}$. At the same time, we adapt the parameters of the encoder to produce a latent embedding that is indistinguishable to the discriminator. This results in matching latent distribution over $z_t^{\mathrm{src}}$ and $z_t^{\mathrm{tgt}}$. Following the Wasserstein GAN (Arjovsky et al., 2017) formulation, we express our distribution matching loss as

$$\mathcal{J}_{\mathrm{adv}} = \mathbb{E}_{P_{\bar{\pi}}(\mathcal{M}_{\mathrm{src}})}\left[D\left(\bar{F}(o_t^{\mathrm{src}})\right)\right] + \mathbb{E}_{P_{\bar{\pi}}(\mathcal{M}_{\mathrm{tgt}})}\left[1 - D\left(F(o_t^{\mathrm{tgt}})\right)\right], \tag{3}$$

where $\bar{F}$ indicates that the weights of the network are frozen. The encoder tries to minimize this objective while the discriminator acts as an adversary and seeks to maximize it, resulting in a GAN-like minimax game.

### 4.2 DYNAMICS CONSISTENCY

Theoretically, the use of adversarial training can result in an encoder that maps source and target domain observations to latent embeddings that have identical distributions. However, it is possible for a sufficiently expressive encoder to map the same target observations to a random perturbation of observations in the source domain while still matching the distributions (Zhu et al., 2017). To mitigate this, we take advantage of the shared underlying dynamics as a feature-rich and informative source of mutual information between the two domains.

---

**Algorithm 1** Invariance through Inference

---

**Require:** Pretrained encoder $F$, discriminator $D$, buffers $\mathcal{B}^*_{\text{src}}$ and $\mathcal{B}_{\text{tgt}}$, inverse dynamics network parameterized with $\psi_{\text{inv}}$

    **for** i=1:N **do**                                                        ▷ Pre-fill buffers
        Sample $z_t, a_t, z_{t+1} \sim P_{\bar{\pi}}(\mathcal{M}_{\text{src}}; F)$
        Sample $o_t, a_t, o_{t+1} \sim P_{\bar{\pi}}(\mathcal{M}_{\text{tgt}})$
        $\mathcal{B}^*_{\text{src}} \leftarrow \mathcal{B}^*_{\text{src}} \cup (z_t, a_t, z_{t+1})$
        $\mathcal{B}_{\text{tgt}} \leftarrow \mathcal{B}_{\text{tgt}} \cup (o_t, a_t, o_{t+1})$
    **for** $i = 1 : T_{\text{dynamics}}$ **do**                                    ▷ Pretrain dynamics networks
        Sample $z_t, a_t, z_{t+1} \sim \mathcal{B}^*_{\text{src}}$
        $g_{\text{inv}} \leftarrow \nabla_{\psi_{\text{inv}}} \mathcal{L}_{\text{dyn}}(z_t, z_{t+1}, a_t; \psi_{\text{inv}})$
        $\psi_{\text{inv}} \leftarrow \text{Optimizer}(\psi_{\text{inv}}, g_{\text{inv}})$
    **for** $i = 1 : T_{\text{adapt}}$ **do**                                         ▷ Adaptation main loop
        Sample $z^{\text{src}}_t, a^{\text{src}}_t, z^{\text{src}}_{t+1} \sim \mathcal{B}^*_{\text{src}}$
        Sample $o^{\text{tgt}}_t, a^{\text{tgt}}_t, o^{\text{tgt}}_{t+1} \sim \mathcal{B}_{\text{tgt}}$
        $g_D \leftarrow \nabla_{\theta_D} \left[ D(z^{\text{src}}_t) + (1 - D(F(o^{\text{tgt}}_t))) \right]$     ▷ Distribution matching (adversarial) loss
        $g_{F_1} \leftarrow \nabla_{\theta_F} \left[ D(z^{\text{src}}_t) + (1 - D(F(o^{\text{tgt}}_t))) \right]$
        $g_{F_2} \leftarrow \nabla_{\theta_F} \mathcal{L}_{\text{dyn}}(F(o^{\text{tgt}}_t), F(o^{\text{tgt}}_{t+1}), a_t; \psi^*_{\text{inv}})$     ▷ Dynamics consistency loss
        $\theta_D \leftarrow \text{Optimizer}(\theta_D, g_D)$
        $\theta_F \leftarrow \text{Optimizer}(\theta_F, g_{F_1}, g_{F_2})$

---

Specifically, let $C_{\text{inv}}(z_t, z_{t+1}; \psi_{\text{inv}})$ be the inverse dynamics network that predicts the action $a_t$ associated with the transition from $z_t$ to $z_{t+1}$, and $C_{\text{fwd}}(z_t, a_t; \psi_{\text{fwd}})$ be the forward dynamics network that predicts the next latent $z_{t+1}$ from $z_t$ and $a_t$. We then define the *dynamics consistency loss* as the error in the inverse dynamics predictions

$$\mathcal{L}_{\text{dyn}}(z_t, z_{t+1}, a_t; \psi_{\text{fwd}}, \psi_{\text{inv}}) = \|C_{\text{fwd}}(z_t, a_t; \psi_{\text{fwd}}) - z_{t+1}\|^2_2 + \|C_{\text{inv}}(z_t, z_{t+1}; \psi_{\text{inv}}) - a_t\|^2_2. \quad (4)$$

Most importantly, we pretrain the inverse dynamics network on transitions sampled from the source domain $\mathcal{B}_{\text{src}} \sim P_{\bar{\pi}}(\mathcal{M}_{\text{src}})$ (Eqn. 1a) with observations encoded with $F$

$$\psi^*_{\text{fwd}}, \psi^*_{\text{inv}} = \underset{\psi_{\text{fwd}}, \psi_{\text{inv}}}{\arg\min} \mathbb{E}_{P_{\bar{\pi}}(\mathcal{M}_{\text{src}}; F)} \left[ \mathcal{L}_{\text{dyn}}(z^{\text{src}}_t, z^{\text{src}}_{t+1}, a^{\text{src}}_t; \psi_{\text{fwd}}, \psi_{\text{inv}}) \right], \quad (5)$$

where $P_{\bar{\pi}}(\mathcal{M}; F)$ is the distribution over latent trajectories. Under the assumption that the underlying dynamics are shared by the domains, we then utilize these learned dynamics models to further encourage the latent embeddings for the target domain to align with those of the source domain. Specifically, during adaptation, we freeze the weights of the forward and inverse dynamics network, and minimize the following dynamics consistency loss with respect to encoder $F$ using transitions sampled from the target domain $\mathcal{B}_{\text{tgt}} \sim P_{\bar{\pi}}(\mathcal{M}_{\text{tgt}})$ (Eqn. 1b)

$$\mathcal{J}_{\text{dyn}} = \mathbb{E}_{P_{\bar{\pi}}(\mathcal{M}_{\text{tgt}}; F)} \left[ \mathcal{L}_{\text{dyn}}(z^{\text{tgt}}_t, z^{\text{tgt}}_{t+1}, a^{\text{tgt}}_t; \psi^*_{\text{fwd}}, \psi^*_{\text{inv}}) \right]. \quad (6)$$

### 4.3 JOINT OBJECTIVE

We adapt our encoder by minimizing a loss that combines both the adversarial loss $\mathcal{J}_{\text{adv}}$ (Eqn. 3) and the dynamics consistency loss $\mathcal{J}_{\text{dyn}}$ (Eqn. 6). Specifically, we solve for the parameters of the encoder through the following minimax objective

$$\underset{F}{\arg\min} \, \underset{D}{\arg\max} \left[ \mathcal{J}_{\text{adv}} + \mathcal{J}_{\text{dyn}} \right]. \quad (7)$$

This adaptation objective is effectively maximizing the mutual information between the prototypical representation kept from during training, and the adapted representations of the new observations. Algorithm 1 summarizes the whole procedures to sample trajectory, pretrain dynamics networks and adapt encoder via the objective.

In summary, in contrast to those methods that produce invariance through interpolation, our adaptation objective produce invariance by making statistical inference in the latent space. For this reason we refer to the derived method as Invariance through Inference.

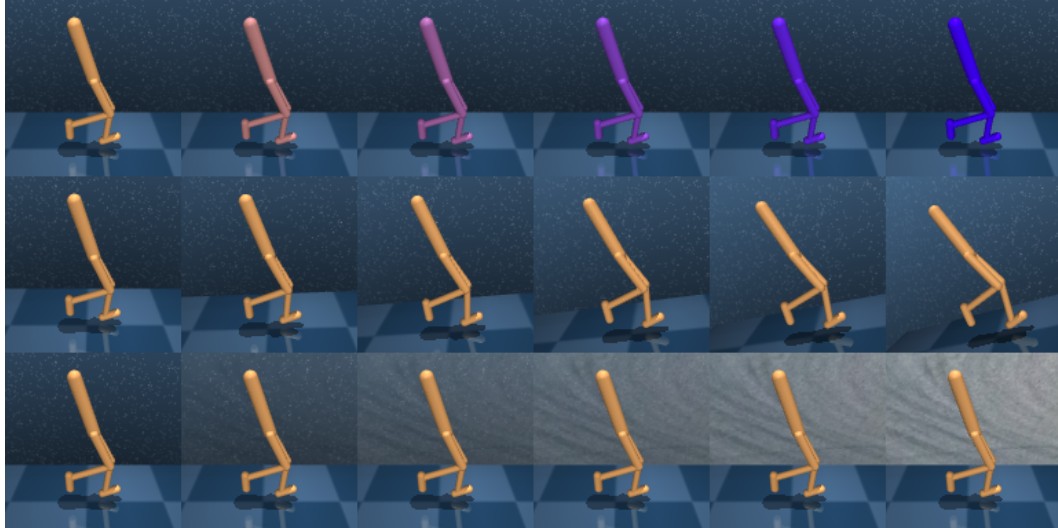

Figure 3: Samples from the modified DistractingCS with intensities increasing from (left-most column) zero to (right-most column) one for the (top row) color, (middle row) camera pose, and (bottom row) background distractions.

## 5 EXPERIMENTS

We want to understand the impact of test-time inference on an agent's ability to perform out-of-distribution generalization. This section will compare Invariance through Inference with two state-of-the-art methods for data-augmented reinforcement learning: SVEA (Hansen et al., 2021) and DrQ-v2 (Yarats et al., 2021). Recall from the introduction that these methods vie for increased generalization capabilities by expanding the support of the training distribution. We expect this form of generalization to be less performant than unsupervised adaptation at test time. In our experiments, we will also compare against Policy Adaptation during Deployment (PAD) (Hansen et al., 2020), a baseline that, like our method, adapts the policy without access to the reward at test time.

To further probe the generalization abilities of test-time inference, we conduct an experiment wherein we vary the intensity of environmental distractions. The upshot here is that test-time inference significantly increases the robustness of our policy to distractions during deployment. We will conclude with some general discussion and remarks regarding the design tradeoffs involved in test-time inference.

**Setup** We conduct experiments using nine domains from the DeepMind Control Suite (DMC) (Tassa et al., 2018). We use DMC as the training (source) environment and the Distracting Control Suite (DistractingCS) (Stone et al., 2021) as the distracted test (target) environment. DistractingCS adds three types of distractions to the DeepMind Control Suite in the form of changes to the background image, deviations in color, and changes to the camera pose relative to training.

**Modifications to DistractingCS** The default configuration of DistractingCS changes distractions at the start of every episode (e.g., different background images are used at every episode). However, we are interested in measuring an agent's ability to perform inference across several episodes on the same target environment. Thus, we modify DistractingCS to sample a distraction once in the beginning of learning, and then use the same distraction across all learning epochs. This also ensures consistent evaluation across algorithms. In accordance with this change, we also modify the intensity benchmark from DistractingCS. In our experiments, intensity measures the deviation between an environment distraction and the train environment's default value. For example, intensity may measure how far the distracting color is from the default. Finally, we modify the environments to only apply a single distraction during testing (rather than all three) in order to better understand the impact of each type of distraction on overall performance. Figure 3 shows an example of distractions across intensities on Walker-walk domain.

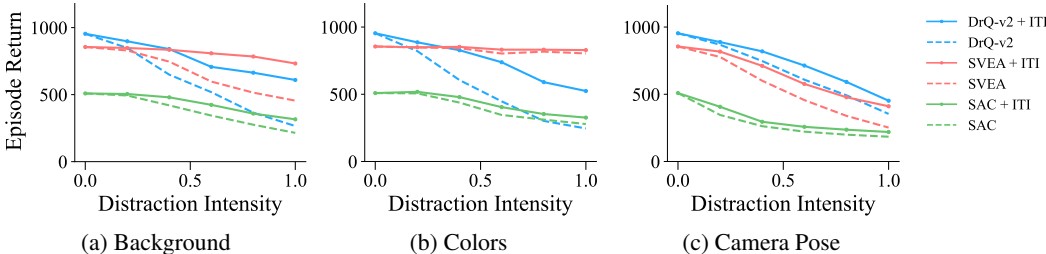

Figure 4: The gain of applying Invariance through Inference to various distracted baselines. Dashed lines denote the performance of the baseline agent in the target environment (i.e., zero-shot), while solid lines represent the performance gains of the base agent with ITI (our method).

## 5.1 Invariance through Inference On DMControl

This section studies the impact of test-time adaptation on DMControl performance. We begin by pretraining soft actor-critic (SAC) (Haarnoja et al., 2018), SVEA, and DrQ-v2 in a non-distracting training-time environment. After training, we evaluate the learned policies on test environments with distractions of various intensities. This evaluation is zero-shot, i.e., there is no additional training in the test environment.

Table 1: Episode return in the target (test) environments (mean and standard deviation) before (zero-shot) and after (+ITI) adaptation for SAC, SVEA, and DrQ-v2 with background distraction at an intensity setting of 1.0. The performance of each baseline in the source (training) environments can be found in the Appendix.

| Domain | SAC | | SVEA | | DrQ-v2 | |
| --- | --- | --- | --- | --- | --- | --- |
| | Zero-shot | +ITI | Zero-shot | +ITI | Zero-shot | +ITI |
| ball_in_cup-catch | $115 \pm 50$ | $\mathbf{227 \pm 222}$ | $490 \pm 376$ | $\mathbf{987 \pm 27}$ | $88 \pm 39$ | $\mathbf{386 \pm 425}$ |
| cartpole-balance | $434 \pm 275$ | $\mathbf{585 \pm 295}$ | $446 \pm 330$ | $\mathbf{627 \pm 258}$ | $273 \pm 107$ | $\mathbf{322 \pm 117}$ |
| cartpole-swingup | $182 \pm 147$ | $\mathbf{369 \pm 243}$ | $269 \pm 365$ | $\mathbf{612 \pm 213}$ | $82 \pm 35$ | $\mathbf{247 \pm 136}$ |
| cheetah-run | $169 \pm 65$ | $\mathbf{248 \pm 53}$ | $317 \pm 137$ | $\mathbf{378 \pm 55}$ | $100 \pm 88$ | $\mathbf{393 \pm 125}$ |
| finger-spin | $113 \pm 162$ | $\mathbf{192 \pm 196}$ | $391 \pm 467$ | $\mathbf{943 \pm 54}$ | $207 \pm 328$ | $\mathbf{769 \pm 206}$ |
| finger-turn_easy | $\mathbf{163 \pm 99}$ | $146 \pm 33$ | $278 \pm 180$ | $\mathbf{491 \pm 343}$ | $268 \pm 241$ | $\mathbf{914 \pm 44}$ |
| reacher-easy | $179 \pm 65$ | $\mathbf{381 \pm 76}$ | $75 \pm 77$ | $\mathbf{624 \pm 305}$ | $58 \pm 32$ | $\mathbf{685 \pm 211}$ |
| walker-stand | $330 \pm 118$ | $\mathbf{364 \pm 115}$ | $917 \pm 138$ | $\mathbf{999 \pm 12}$ | $630 \pm 197$ | $\mathbf{868 \pm 151}$ |
| walker-walk | $242 \pm 142$ | $\mathbf{291 \pm 134}$ | $866 \pm 45$ | $\mathbf{924 \pm 45}$ | $326 \pm 195$ | $\mathbf{770 \pm 140}$ |

Table 1 presents the results for the different DistractingCS domains in the presence of background distractions with an intensity level of 1.0. Specifically, we compare the test-time performance of SAC, SVEA, and DrQ-v2 in each domain with the episode rewards that we achieve when using our Invariance through Inference algorithm to adapt the encoder. The baseline algorithms employ image augmentation, which provides some robustness to variations at test time. Even then, however, we find that Invariance through Inference adaptation improves the test-time generalization of all three baseline policies in most domains, often resulting in significant performance gains. In cases where Invariance through Inference does not improve performance, the resulting reward is comparable to the baseline policy, i.e., Invariance through Inference does not result in a performance degradation.

Figure 4 visualizes the performance of the different methods, averaged over the set of DistractingCS domains, as a function of the intensity of the distractions. Since the baseline methods are trained with image augmentation, they do exhibit some robustness to distraction. However, we see this robustness rapidly diminishes as the distraction intensity increases. In particular, large changes to camera pose or the image background proved challenging for standard augmentation procedures. Comparatively, Invariance through Inference makes it much smoother and slower degradation of performance. This supports our hypothesis that adaptation powered by unsupervised learning can significantly widen the generalization abilities of learning algorithms.

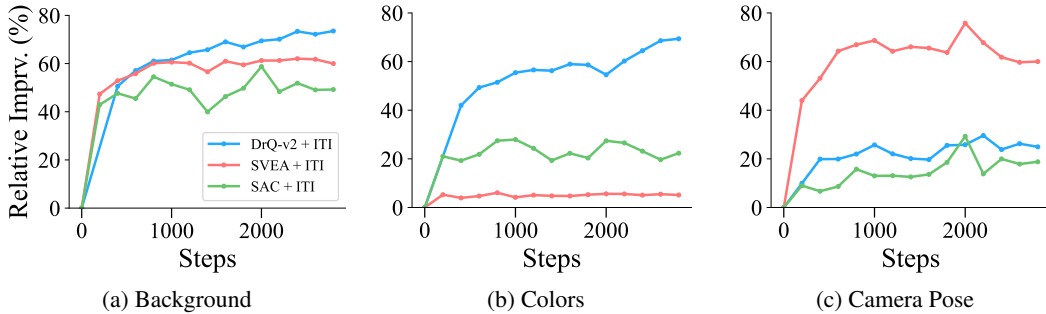

Figure 5: Relative improvement (compared to zero-shot) as a function of adaptation steps when applying ITI to different baseline policies. As in Figure 4, each point represents the mean over nine domains and five random seeds. The results correspond to an intensity value of 1.0.

Table 2: Comparison against PAD

| Distraction | Zero-shot | +PAD | +ITI |
|---|---|---|---|
| None | $835 \pm 230$ | — | — |
| Background | $213 \pm 247$ | $279 \pm 271$ | $\mathbf{425 \pm 292}$ |
| Colors | $230 \pm 263$ | $271 \pm 300$ | $\mathbf{402 \pm 339}$ |
| Camera Pose | $319 \pm 265$ | $326 \pm 259$ | $\mathbf{412 \pm 275}$ |

## 5.2 COMPARISONS WITH PAD

Similar to our approach, PAD pretrains the agent in a clean environment, and then adapts the agent via unsupervised transfer, without assuming access to the target environment's reward function (Hansen et al., 2020). To evaluate the robustness of PAD to distractions, we consider DistractingCS with a fixed distraction intensity of 1.0. Table 2 compares the performance as the difference between the episode returns before and after adaptation along with the episode returns in the clean environment. It should be noted that PAD requires to pretrain a policy along with inverse dynamics prediction objective. Thus we trained SAC with the auxiliary objective specifically for this experiment.

Across all environments, we see that PAD struggles to adapt to distractions at test time. We suspect this instability is caused by the large deviations in the latent variable distribution as a result of changes in the target environment. In particular, we posit that the signal from PAD's inverse dynamics head does not encourage the latent train and test distributions to match, which is a feature specifically baked into Invariance through Inference.

## 5.3 FURTHER DISCUSSION

Table 3: Ablations with variants of Invariance through Inference that remove inverse, forward dynamics, or adversarial objectives. DrQ-v2 is used as a pretrained policy. We compute episode returns from nine domains and five random seeds, and the results correspond to an intensity value of 1.0.

| Distraction | Zero-shot | +ITI | +ITI w/o dyn. | +ITI w/o adv. |
|---|---|---|---|---|
| Background | $228 \pm 232$ | $602 \pm 300$ | $615 \pm 289$ | $176 \pm 221$ |
| Colors | $234 \pm 245$ | $536 \pm 320$ | $534 \pm 327$ | $117 \pm 96$ |
| Camera Pose | $345 \pm 287$ | $417 \pm 284$ | $407 \pm 272$ | $208 \pm 235$ |

**Ablation Studies** In order to better understand the contribution of the different objectives to test-time generalization, we perform a series of ablations in which we omit either the dynamics consistency or the adversarial objectives. In these experiments, we use a pretrained DrQ-v2 network

for the algorithm's base policy, and then perform adaptation across all distractions with an intensity value of 1.0. The results in Table 3 show that the adversarial training is critical to adapt the latent representation in the target domain. Performing adaptation using only the dynamics consistency objective, i.e., $\arg\min_F \mathcal{J}_{\text{dyn}}$ (Eqn. 6) results in a significant decrease in performance. We theorize that the dynamics consistency objective helps to refine the local consistency in the latent space when the space in the target domain is close to that of the source domain. If the latent spaces significantly differ, however, we suspect that the gradients may negatively affect convergence.

Compared to the adversarial objective, ablating the dynamics consistency objective has surprisingly little effect on test-time generalization. It may be that the local structure of the latent space is preserved despite the distractions, which then diminishes the net effect of the dynamics consistency objective. For example, one can think of the original points in the latent space simply being shifted and affine-transformed. In this case, solely matching the distributions can effectively undo the transformations, resulting in a representation that is consistent with the original, without the need to explicitly align local structure. We suspect that this may be the case in the domains that we considered, and thus the effect of dynamics consistency objective gets obfuscated.

**Pre-Filling the Replay Buffer**  In our experiments, we pre-filled both the original latent buffer and the target observation buffer with data collected via random exploration. There is nothing particularly special about this choice, random exploration may be subbed out with any exploration strategy. One interesting strategy is to pre-fill the buffer with an exploration strategy that relies on unsupervised pretraining. Given the recent success of these strategies on improving exploration, it seems likely this could further improve performance. We leave this for future work.

## 6 CLOSING REMARKS

We introduced Invariance through Inference, a new method for leveraging unsupervised data to improve the test-time adaptation of reinforcement learning systems. Empirical results demonstrate that as discrepancies between training and deployment environments become more intense, Invariance through Inference is more capable of adaptation than data augmentation techniques. The problem of test-time adaptation in visual reinforcement learning using unsupervised test-time trajectories is relatively new, but has thus far shown great promise.

Future work in this area might develop better techniques for the initial exploration phase. Many of the environments we considered were quite reversible, with a small intrinsic dimensionality. Expanding this class of methods to work on longer horizon multi-stage tasks is an intriguing possibility.

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

## A  APPENDIX

The following provide a more detailed experimental evaluation of Invariance through Inference on the Distracting Control Suite.

## B  PERFORMANCE IN ORIGINAL DOMAINS

Table 4 presents the average reward for the baseline SAC, SVEA, and DrQ-v2 on the non-distracted source domains. The method labeled SAC+Inv denotes a soft actor-critic agent that is trained using inverse dynamics as an additional auxiliary objective that is only used in making a comparison against PAD.

Table 4: Performance in the original (clean) domains.

| Domain | SAC | SAC+Inv | SVEA | DrQ-v2 |
|---|---|---|---|---|
| ball_in_cup-catch | $452 \pm 303$ | $999 \pm 7$ | $1007 \pm 4$ | $1007 \pm 3$ |
| cartpole-balance | $1022 \pm 8$ | $988 \pm 26$ | $996 \pm 21$ | $969 \pm 123$ |
| cartpole-swingup | $735 \pm 167$ | $885 \pm 24$ | $892 \pm 15$ | $874 \pm 21$ |
| cheetah-run | $309 \pm 26$ | $415 \pm 59$ | $448 \pm 105$ | $897 \pm 45$ |
| finger-spin | $615 \pm 63$ | $971 \pm 64$ | $1000 \pm 37$ | $997 \pm 36$ |
| finger-turn_easy | $138 \pm 28$ | $658 \pm 141$ | $539 \pm 317$ | $945 \pm 46$ |
| reacher-easy | $381 \pm 39$ | $723 \pm 383$ | $812 \pm 293$ | $988 \pm 28$ |
| walker-stand | $438 \pm 111$ | $997 \pm 6$ | $1006 \pm 4$ | $996 \pm 31$ |
| walker-walk | $393 \pm 117$ | $915 \pm 22$ | $964 \pm 34$ | $980 \pm 16$ |

## C  ABLATIONS

Tables 5, 6, and 7 provide a per-domain ablation summary for background, color, and camera pose distractions, respectively. As with the results in Table 3, we use DrQ-v2 as the pretrained policy and present the mean reward and standard deviation for five random seeds.

Table 5: Ablation with Background distraction

| Domain | Zero-shot | +ITI | +ITI w/o inv., fwd. | +ITI w/o inv. | +ITI w/o fwd. | +ITI w/o adv. |
|---|---|---|---|---|---|---|
| walker-walk | $326 \pm 196$ | $749 \pm 133$ | $778 \pm 142$ | $777 \pm 145$ | $768 \pm 146$ | $268 \pm 367$ |
| walker-stand | $623 \pm 233$ | $866 \pm 153$ | $883 \pm 110$ | $859 \pm 144$ | $873 \pm 129$ | $402 \pm 347$ |
| cartpole-swingup | $82 \pm 35$ | $231 \pm 128$ | $395 \pm 218$ | $288 \pm 163$ | $246 \pm 152$ | $117 \pm 43$ |
| ball_in_cup-catch | $88 \pm 39$ | $394 \pm 387$ | $381 \pm 416$ | $400 \pm 414$ | $383 \pm 381$ | $93 \pm 42$ |
| finger-spin | $208 \pm 327$ | $783 \pm 214$ | $742 \pm 190$ | $772 \pm 225$ | $769 \pm 223$ | $74 \pm 160$ |
| reacher-easy | $98 \pm 93$ | $726 \pm 149$ | $713 \pm 111$ | $732 \pm 104$ | $694 \pm 169$ | $91 \pm 54$ |
| cheetah-run | $98 \pm 90$ | $411 \pm 191$ | $397 \pm 152$ | $398 \pm 147$ | $419 \pm 157$ | $12 \pm 19$ |
| cartpole-balance | $271 \pm 101$ | $336 \pm 126$ | $315 \pm 98$ | $367 \pm 84$ | $297 \pm 121$ | $264 \pm 80$ |
| finger-turn_easy | $261 \pm 244$ | $920 \pm 41$ | $929 \pm 57$ | $899 \pm 52$ | $909 \pm 74$ | $256 \pm 264$ |

## D  ARCHITECTURES

This section describes the architectures of encoder $F$, discriminator $D$, inverse dynamics $C_{\text{inv}}$ and forward dynamics $C_{\text{fwd}}$. Encoder architectures follow the originally presented design choices of each base policy except for DrQ-v2. We take the part of the original network that produces a latent for the actor, and use it as an encoder $F$. In all of SAC (of the version used in (Hansen et al., 2021)), SVEA and PAD, this shared latent is set to have dimension 100. For DrQ-v2, we took the entire network architecture from SVEA. The discriminator consists of a linear layer with hidden dimension 100 followed by Layer Normalization (LN) (Ba et al., 2016) and tanh activation, and a three layer

Table 6: Ablation with Color distraction

| Domain | Zero-shot | +ITI | +ITI w/o inv., fwd. | +ITI w/o inv. | +ITI w/o fwd. | +ITI w/o adv. |
|---|---|---|---|---|---|---|
| walker-walk | $80 \pm 43$ | $481 \pm 313$ | $468 \pm 305$ | $495 \pm 330$ | $472 \pm 319$ | $26 \pm 5$ |
| walker-stand | $278 \pm 150$ | $543 \pm 231$ | $603 \pm 283$ | $571 \pm 259$ | $548 \pm 268$ | $145 \pm 31$ |
| cartpole-swingup | $152 \pm 84$ | $552 \pm 377$ | $520 \pm 359$ | $507 \pm 339$ | $434 \pm 395$ | $94 \pm 56$ |
| ball_in_cup-catch | $239 \pm 374$ | $812 \pm 225$ | $857 \pm 182$ | $840 \pm 251$ | $843 \pm 196$ | $131 \pm 46$ |
| finger-spin | $349 \pm 354$ | $612 \pm 345$ | $561 \pm 376$ | $591 \pm 351$ | $603 \pm 358$ | $143 \pm 172$ |
| reacher-easy | $138 \pm 135$ | $490 \pm 416$ | $486 \pm 392$ | $486 \pm 362$ | $445 \pm 405$ | $140 \pm 90$ |
| cheetah-run | $193 \pm 188$ | $422 \pm 282$ | $416 \pm 293$ | $421 \pm 277$ | $406 \pm 285$ | $4 \pm 2$ |
| cartpole-balance | $481 \pm 351$ | $602 \pm 366$ | $576 \pm 359$ | $593 \pm 351$ | $583 \pm 349$ | $216 \pm 94$ |
| finger-turn_easy | $194 \pm 200$ | $313 \pm 290$ | $323 \pm 337$ | $297 \pm 316$ | $304 \pm 323$ | $170 \pm 57$ |

Table 7: Ablation with Camera Pose distraction

| Domain | Zero-shot | +ITI | +ITI w/o inv., fwd. | +ITI w/o inv. | +ITI w/o fwd. | +ITI w/o adv. |
|---|---|---|---|---|---|---|
| walker-walk | $293 \pm 195$ | $375 \pm 154$ | $369 \pm 168$ | $389 \pm 161$ | $366 \pm 164$ | $63 \pm 71$ |
| walker-stand | $621 \pm 202$ | $704 \pm 105$ | $617 \pm 155$ | $661 \pm 146$ | $679 \pm 93$ | $330 \pm 252$ |
| cartpole-swingup | $286 \pm 51$ | $234 \pm 123$ | $211 \pm 122$ | $241 \pm 140$ | $281 \pm 42$ | $172 \pm 132$ |
| ball_in_cup-catch | $327 \pm 227$ | $462 \pm 307$ | $429 \pm 321$ | $566 \pm 314$ | $400 \pm 337$ | $199 \pm 169$ |
| finger-spin | $29 \pm 25$ | $252 \pm 216$ | $222 \pm 209$ | $275 \pm 194$ | $251 \pm 206$ | $32 \pm 63$ |
| reacher-easy | $917 \pm 101$ | $933 \pm 47$ | $925 \pm 60$ | $950 \pm 72$ | $940 \pm 90$ | $732 \pm 159$ |
| cheetah-run | $55 \pm 20$ | $142 \pm 57$ | $154 \pm 90$ | $141 \pm 74$ | $144 \pm 61$ | $24 \pm 28$ |
| cartpole-balance | $288 \pm 46$ | $307 \pm 100$ | $375 \pm 83$ | $379 \pm 116$ | $380 \pm 64$ | $217 \pm 47$ |
| finger-turn_easy | $287 \pm 89$ | $346 \pm 219$ | $359 \pm 122$ | $377 \pm 200$ | $421 \pm 180$ | $160 \pm 97$ |

multi-layer perceptron (MLP) with and ReLU activations. Inverse dynamics network $C_{\text{inv}}$ is five layer MLP and ReLU activations. It takes the concatenated latents as its input. Forward dynamics network $C_{\text{fwd}}$ takes action and latent, and encode them separately followed by concatenation and further layers. The action is fed it to a linear layer followed by LN, another linear layer and ReLU with hidden dimension 100. The latent is fed to three layer MLP where the first activation is LN and others are ReLU. The both encoded inputs are concatenated and fed to 4 layer MLP with ReLU activations followed by LN and tanh activation. In all layers except for those explicitly mentioned, hidden dimension is set to $1,024$.

# E  HYPERPARAMETERS

This section details the hyperparameter settings that were used for the experimental evaluation. The hyperparameters relevant to pretraining dynamics network are listed in Table 8, and those relevant to adaptation are listed in Table 9. For the base policies, we followed the hyperparameters and architecture choices on their original paper. Those information on SAC and SVEA can be found in (Hansen et al., 2021), and PAD in (Hansen et al., 2020). The dimension of the latent $z_t$ differs depending on the encoder of the base policy, but all of the encoders in our experiments produce a latent with dimension 100.

Table 8: Hyperparameters for dynamics pretraining

| Hyperparameter | Value |
|---|---|
| Steps ($T_{\text{dyn}}$) | $100,000$ |
| Batch size | $256$ |
| Optimizer | RMSProp($\alpha = 0.99, \epsilon = 1.0 \times 10^{-8}$) |
| Learning rate (forward dynamics) | $0.001$ |
| Learning rate (inverse dynamics) | $0.001$ |

Table 9: Hyperparameters for adaptation

| Hyperparameter | Value |
|---|---|
| Capacity of buffers ($N_{\text{buf}}$) | $1,000,000$ |
| Batch size | 256 |
| Discriminator updates per step | 5 |
| Gradient clipping | 0.01 |
| Optimizer | RMSProp($\alpha = 0.99, \epsilon = 1.0 \times 10^{-8}$) |
| Learning rate (encoder) | $1.0 \times 10^{-4}$ (for DrQ-v2) |
|  | $1.0 \times 10^{-5}$ (otherwise) |
| Learning rate (discriminator) | $1.0 \times 10^{-4}$ (for DrQ-v2) |
|  | $1.0 \times 10^{-5}$ (otherwise) |
| Learning rate (inverse dynamics) | $1.0 \times 10^{-6}$ |

