# OpenReview forum: "Invariance Through Inference"
_ICLR.cc/2022/Conference — ICLR 2022 Submitted_

### Official Review · Reviewer_HvyM · 2021-10-29

**Correctness:** 4
**Technical Novelty And Significance:** 3
**Empirical Novelty And Significance:** 3
**Recommendation:** 6
**Confidence:** 4

**Main Review:**

I quite like the aim of the proposed method, and the results are impressive.  The fundamental principle of using the discriminator to obtain invariant latent representation of an out-of distribution state space of the MDP seems like a very sensible and practical idea.  The only problem with the paper is that is very high level and I can't quite figure out how it all works and comes together.  Even with Figure 3 and Algorithm overview in section 4.2, it's not clear what happens when.  Seems there are different phases to training...but I can't quite understand what happens when.  When is P_\phi fixed and when it is trainable?  What is a_t in Equation 2 when training on MDP_test?  What's z and \tilde{z} in 3?  Do D_{non_dist} and D_{distr} refer to data from MDP_{train} and MDP_{test}?  How big does the source buffer storing training data need to be?  Does it store all training data?

**Summary Of The Paper:**

The paper proposes a method for learning invariant latent representation of the observations from MDP processes that share some aspects of their dynamics but differ in states.  This is useful for generlising reinforcement learning agents to wider range of variability of test conditions (i.e. test data can be out of distribution of the train data).  The invariant latent representation is derived through minimisation of mutual information between latent encoding of the in-distribution and the out-of-distribution experiences by "fooling" a GAN-like discriminator tasked at differentiating the two.

**Summary Of The Review:**

I don't think it would require a huge amount of work, but a bit more meticulous explanation of the loss functions and the training process is needed.  At the moment it seems that only those intimately familiar with the method and code (that is not provided) would be able to implement it.  The direction seems good, results are very encouraging, just need to flesh out the actual method a bit more, otherwise it feels there isn't enough information to reproduce.

---

> ### Author Response · Authors · 2021-11-12
> **Thank you for your feedback & comments!**
>
> Thank you so much for your thoughtful review! We apologize for the hasty explanation. We take reproducibility and clarity seriously. Here are two things we are working on at the moment:
>
> 1. We have just uploaded a clean-up version of our experimental code base as part of the supplementary material for your review. We plan to release this to public as soon as our work is accepted.
>
> 2. We have completely re-written the method section, and are restructuring the results section. We should be able to finish this in two phases over the next couple of days and will provide an updated draft as soon as possible.
>
> **Other quick responses**
>
> **What is a_t in Equation 2 when training on MDP_test?**
> Our assumption for MDP_test is that the agent cannot access its reward. The agent can still access the current observation and the next observation given an action. We will clarify the narrative in the updated draft.
>
> **How big does the source buffer storing training data need to be?**
> Both source buffer and target buffer store 1 million transitions. As each episode has 1k steps, it corresponds to storing 1k episodes. We will put these hyperparameters in the supplementary section.
>
> Updating a paper is going to take us a few days, and we sincerely hope that you can take a look and update your evaluation based on it. Thank you again for your thoughtful review!

---

> ### Author Response · Authors · 2021-11-18
> **We have significantly updated section 1 - 4 on the paper.**
>
> Dear reviewer HvyM,
>
> Thank you so much again for giving us thorough feedback and comments.
>
> Following your feedback, we have significantly updated sections 1 - 4 on the paper, restructuring the text to clarify the ideas and narratives. We have also fixed wrong or inconsistent notations that you have kindly pointed out altogether.
>
> We believe this update addresses most of the concerns and flaws you mentioned.
> Especially we added an algorithm box, and largely updated the description of our method. We hope you find it easier to understand the exact procedures in the proposed method. But please feel free to let us know if anything still sounds unclear.
>
> We put so much time into updating the paper following your kind feedback. So we would sincerely appreciate it if you could spend some time going through the updated sections 1 - 4, and edit the score accordingly if you believe the paper got improved by addressing your concerns.
>
> NOTE: we have not edited the Experiment section at this point yet. But we are working very hard to update some of the experiment results right now. We should be able to reflect those results in a few days.

---

### Official Review · Reviewer_X3P2 · 2021-11-02

**Correctness:** 2
**Technical Novelty And Significance:** 3
**Empirical Novelty And Significance:** 3
**Recommendation:** 5
**Confidence:** 4

**Main Review:**

The paper devotes a lot of space to motivating the setup and justifying its value. I do believe that this setup is relevant in practice in robotics applications. There are indeed many cases in which sim2real faces generalization problems. Unfortunately, the value of this approach is not actually demonstrated in a sim2real application, but in transfer between simulated applications.

While the goal of the paper seems very worthy, and the proposed algorithm seems to provide empirical benefits, the theoretical presentation and the justification of the approach are severely lacking, and I would say incorrect. In particular:
- Why is a distinction between S and O even made? The presentation goes beyond the standard RL presentation making a distinction between the state and the observation, and makes a point of saying that the observation doesn't uniquely identify the state. And that the transition function P is a function of states (and actions) and not observations. To then ignore such distinction and make the transition function only a function of the observations, negating the explanation about partial observability.
- Why is Eq. (5) just adding the two losses? Is this dimensionally correct? How do we know they are balanced? It seems that the relative scaling could be arbitrary by simply changing the scale of the actions.
- What's \tilde{z} in Eq. (3)? This is never defined.
- In Eq. (4),  \bar{G} is said to be the frozen generator. However, this frozen generator is put inside the expectation over "outside of the distribution samples". Isn't the frozen generator used for the training samples, therefore inside the distribution? There's no proper definition of what "distr" and "non-distr" means.
- At the beginning of section 4.1, the case is made for a single generator being used both for training and testing data, and that this generator should ignore the distractors. However, Eq. (4) and the rest of the paper uses different generators for training and testing.
- Figure 4.2 Are you sure this is a D? Shouldn't this be a P_\phi?
- Most importantly, the use of the GAN to match the distributions over z is completely unjustified, see more below.

The need for the GAN: It is unclear what the GAN portion of the architecture is bringing to the table. Since the inverse dynamics head is locked and learned on the training data, making it work on the test data should result in a z that aligns well with the latent space discovered at training time. The paper contains an ablation section showing that removing the GAN makes the system not work at all. How is this possible? The authors hypothesize that "the inverse dynamics head has a tendency to simply memorize dynamics in the training and testing domains without collapsing the commonalities". But the authors had said that, after training the inverse dynamics head on the training data, this head is frozen and remains locked during adaptation to test data. How can it possibly memorize the test dynamics then? And, even though I don't understand how this overfitting can happen, if it did, shouldn't the solution be regularization of the inverse dynamics network? The account of this architecture is inconsistent and the GAN seems completely unwarranted. Very little detail about the whole process is given and there's not supplementary information or code available.

Minor:
Typo: "training environment the Distracting"
The meaning of the values in Table 1 are unclear. The legend has no information and the text says it's the reward difference between before and after adaptation, but then there's a column for before and another for after, so the numbers themselves are not the difference. What is it?

**Summary Of The Paper:**

In this paper, the transfer of a reinforcement model (RL) setting from an idealized (training) environment to a more realistic environment with distractors in the observations is considered. Instead of augmenting the training environment with more data so as to make the system more resilient to variations and distractors, the system is adapted at test time to be invariant to the specific distractors found in the environment. Experiments in simulation show the benefits of the proposed approach. Crucially, the agent is not able to access any reward data at test time.

**Summary Of The Review:**

This paper tackles an important problem, but contains technical inconsistencies, lacks rigor and provides very little detail about the proposal as a whole. The use of a GAN, a major part of their architecture, is completely unjustified, and we are told that it's necessary because the system doesn't work if we remove it... but there's no explanation as to why. No code is provided, no supplementary details are provided.

Empirical results seem good, but there's little support as to why this is the case.

---

> ### Author Response · Authors · 2021-11-12
> **Thank you for your feedback & comments!**
>
> Thank you so much for thoroughly reviewing the paper! We are terribly sorry for the lack of rigor and the lack of clarity in our presentation of the method. To fix this issue, we are doing these two things at the moment:
>
> 1. We just uploaded a clean-up version of our codebase as the supplementary material. We plan to make this code available on GitHub as soon as this paper is accepted.
>
> 2. Since submission we have already re-written the method section. We are in the process of updating the paper, and will provide you with a more comprehensive update in a few days.
>
> **Other quick responses**
>
> **As with the "GAN" objective** -- we have changed our write-up and now consider this domain adaptation scheme a distribution matching objective in the latent space.
>
> **How to interpret Table 1?**  The values in table 1 are episode returns (score). The table shows how the score gets improved by the adaptation. For example, (Before Adaptation - ITI - Camera) shows the score of the agent deployed in the environment with camera distraction. This corresponds to the zero-shot performance. If we compare that value against (After Adaptation - ITI - Camera), we can see that the score gets improved after running adaptation. (Before Adaptation - ITI - None) and (Before Adaptation - PAD - None) shows the scores in the same (distraction-free) domain as they are trained.
>
> **Why is PAD (adaptation with inverse dynamics) not enough?**
> The inverse modeling objective forces the **local transition between two near-by states** to contain mutual information against the action. This is a local object that does not enforce **global consistency over the overall distribution**. We consider that the distribution matching in latent space via adversarial training encourages the global consistency (i.e., matching marginal distributions of z), and aligning both local and global structure is crucial to make the agent perform well in the target environment. We are now re-writing the text to describe this clearer.
>
> Updating a paper is going to take us a few days, and we sincerely hope that you can take a look and update your evaluation based on it. Thank you again for your thoughtful review!

---

> ### Author Response · Authors · 2021-11-18
> **Response (1/3)**
>
> (This response was edited for better clarifications)
> Dear reviewer X3P2,
>
> Thank you so much again for giving us thorough feedback and comments.
>
> In the following, we address each of your question / comment:
>
> > There are indeed many cases in which sim2real faces generalization problems. Unfortunately, the value of this approach is not actually demonstrated in a sim2real application, but in transfer between simulated applications.
>
> We agree with that. We did not have enough time to work on sim2real experiments for during discussion period, and we plan to add those experiments in the future work.
> However, we want to note that we experimented on nine simulated robotics domain, which is comparably a large number.
> Moreover, since we have planned to add sim2real experiments with robotics manipulation task after the discussion period anyway, we will be able to put those experiments in the camera-ready version if the paper is accepted.
>
> > Why is a distinction between S and O even made? The presentation goes beyond the standard RL presentation making a distinction between the state and the observation, and makes a point of saying that the observation doesn't uniquely identify the state. And that the transition function P is a function of states (and actions) and not observations. To then ignore such distinction and make the transition function only a function of the observations, negating the explanation about partial observability.
>
> We have largely rewritten Problem formulation (Section 3).
> We clarified that we consider a fully-observable setting where its internal state can be completely predicted from an observation (a stack of images in practice) given a good feature extractor.
> We also rewrote the draft so that the dynamics and reward function in an MDP is defined based on states as they are in a common MDP formulation.
>
> > Why is Eq. (5) just adding the two losses? Is this dimensionally correct? How do we know they are balanced? It seems that the relative scaling could be arbitrary by simply changing the scale of the actions.
>
> Originally Eq. 5 was $\mathcal{J}\_\text{d} = \textrm{argmin}\_D \bigg[ \log P(z) + (1 - \log P(\tilde z))  \bigg]$. Now this is moved to Eqn. 7.
> Both of the losses are scalars. We can potentially have a coefficient on one of the terms to balance the losses, however, we can achieve the same by changing learning rates of the corresponding optimizer. Thus we didn't add such a coefficient. As you mentioned, the scale of the action space matter, and one may need to adjust the learning rate according to it in the current formulation. However, in Deep Mind Control suite, the maximum norm of an action is fixed even though its dimension varies across domains. Thus we kept using the same learning rate.
>
> > What's \tilde{z} in Eq. (3)? This is never defined.
>
> We replaced the equation so that it reveals more concrete details:
> Original Eqn. 3: $\mathcal J\_D =  \mathbb{E}\_{o\_t \sim D\_\textrm{non-distr}}\big[ \log D(\bar{G}\_\theta(o\_t)) \big] + \mathbb{E}\_{o\_t \sim D\_\text{distr}}\big[ \log (1 - D(G\_\theta(o\_t))) \big] \bigg]$.
>
> Current Eqn. 3:
>
>  $\mathcal{J}\_\textrm{adv} = \mathbb{E}\_{P_{\bar{\pi}}(\mathcal{M}\_\textrm{src})} \left[ D\left(\bar{F} (o^{\text{src}}\_t)\right) \right] + \mathbb{E}\_{P\_{\bar{\pi}}(\mathcal{M}\_\textrm{tgt}) }\left[1 - D\left(F(o\_t^{\text{tgt}})\right) \right]$
>
> Instead of $z$ and $\tilde{z}$, we now have $\bar{F}(o^{\textrm{src}}\_t)$ and $F(o^\text{tgt}\_t)$, where $F$ is encoder, $\bar{F}$ represents that the weights are frozen, and $o^{\text{src}}\_t$, $o^\text{tgt}\_t$ are obsevations in source domain and target domain.
> We made sure that all the notations are defined.

---

> ### Author Response · Authors · 2021-11-26
> **Response (2/3)**
>
> > In Eq. (4), \bar{G} is said to be the frozen generator. However, this frozen generator is put inside the expectation over "outside of the distribution samples". Isn't the frozen generator used for the training samples, therefore inside the distribution? There's no proper definition of what "distr" and "non-distr" means.
>
> This is the original Eqn. 4 $\mathcal J\_D =  \mathbb{E}\_{o\_t \sim D\_\textrm{non-distr}}\big[ \log D(\bar{G}\_\theta(o\_t)) \big] + \mathbb{E}\_{o\_t \sim D\_\text{distr}}\big[ \log (1 - D(G\_\theta(o\_t))) \big] \bigg],$
>
> We now have the new Eqn. 3 that should be able to clarify your concern:
> $\mathcal{J}\_\textrm{adv} = \mathbb{E}\_{P_{\bar{\pi}}(\mathcal{M}\_\textrm{src})} \left[ D\left(\bar{F} (o^{\text{src}}\_t)\right) \right] + \mathbb{E}\_{P\_{\bar{\pi}}(\mathcal{M}\_\textrm{tgt}) }\left[1 - D\left(F(o\_t^{\text{tgt}})\right) \right]$
>
> The encoder $F$ of a pretrained agent is frozen ($\bar{F}$) and is used to encode observations from source domain $o^\textrm{src}\_t$. In practice, we store encoded transitions: $(\bar{F}(o^t\_\textrm{src}), a\_t, \bar{F}(o^{t+1}\_\textrm{src}))$ to the source buffer $\mathcal{B}^*\_\textrm{src}$ as a preprocessing step, and sample latents from it to compute the first term in Eqn. 3.
> For the second term, observations in target domain $o^\textrm{tgt}\_t$ is fed to encoder and decoder, and the resulting gradients are used to update encoder weights.
>
> We removed "distr" and "non-distr", and introduced source domain $\mathcal{M}\_\textrm{src}$ and target domain $\mathcal{M}\_\textrm{tgt}$.
> As can be seen on Eqn. 1a and 1b, we now use $P\_\bar{\pi}(\mathcal{M}\_\textrm{src})$ and $P\_\bar{\pi}(\mathcal{M}\_\textrm{tgt})$ as distribution over random trajectories in each domain.
>
> > At the beginning of section 4.1, the case is made for a single generator being used both for training and testing data, and that this generator should ignore the distractors. However, Eq. (4) and the rest of the paper uses different generators for training and testing.
>
> This was our fault not to clearly describe how our approach works.
> As is clarified by newly added Algorithm 1, our algorithm first pretrain an agent in a source domain with a standard RL method.
> Then we take the encoder part of the agent and freeze its weight to obtain $\bar{F}$.
> After that we use that encoder to encode observations and store random latent trajectory into *source buffer* $\mathcal{B}\_\textrm{src}^{\*}$. By sampling latents from this source buffer, we compute the first term of current Eqn. 3 as:
>
> $\mathcal{J}\_\textrm{adv} = \mathbb{E}\_{z\_t \sim \mathcal{B}^{\*}\_\textrm{src}(\mathcal{M}\_\textrm{src})} \left[ D(z\_t) \right]$.
>
> On the other hand, the second term of Eqn 3 is computed as:
> $\mathbb{E}\_{o\_t^\textrm{tgt} \sim \mathcal{B}\_\textrm{tgt}} \left[1 - D\left(F(o\_t^{\text{tgt}})\right) \right])$,
>
> where $\mathcal{B}\_\textrm{tgt}$ is the target buffer that contains random trajectory in the target domain.
> In here, the weights of $F$ is initialized to that of the pretrained encoder, and then updated as adaptation goes.
>
> Thus, we only use one encoder, and $\bar{F}$ is only used to fill source buffer $\mathcal{B}^*\_\textrm{src}$ as a preprocess.
>
> > Figure 4.2 Are you sure this is a D? Shouldn't this be a P_\phi?
>
> Thanks for spotting this typo. We have removed Fig. 4 on the updated draft, as this preprocess step becomes clear on the newly edited Section 4.1.
>
> > The need for the GAN: It is unclear what the GAN portion of the architecture is bringing to the table. Since the inverse dynamics head is locked and learned on the training data, making it work on the test data should result in a z that aligns well with the latent space discovered at training time.
>
> It is true that if dynamics consistency loss (inverse + newly added forward dynamics objective) is minimal on target domain, then the latent spaces should be aligned very well.
> However, that does not necessarily mean minimizing the dynamics consistency loss always lead the encoder weights to a good local optimum.
> In fact, we theorize that the dynamics consistency objective helps to **refine the local consistency** in the latent space when the space in the target domain is already close to that of the source domain. If the latent spaces significantly differ, however, we suspect that the gradients may even negatively affect convergence. Thus, aligning the (marginal) distribution of latents via adversarial training is more crucial.
>
> And this aligns well with our empirical results in ablation studies (Section 5.3).

---

> ### Author Response · Authors · 2021-11-26
> **Response (3/3)**
>
> > The authors hypothesize that "the inverse dynamics head has a tendency to simply memorize dynamics in the training and testing domains without collapsing the commonalities". But the authors had said that, after training the inverse dynamics head on the training data, this head is frozen and remains locked during adaptation to test data. How can it possibly memorize the test dynamics then?
>
> As you pointed out, the original phrasing was slightly misleading.
> We have rewritten the discussion around the roles of dynamics consistency objective (i.e., adjusting local consistency in the latent space) and
> adversarial objective (i.e., matching latent distribution to align latent structure globally).
>
> > Very little detail about the whole process is given and there's not supplementary information or code available.
>
> There are updates we made to address this:
> - We uploaded the code as a supplementary material
> - We added the algorighm box (Algorithm 1) to clarify the procedures of our approach
> - We have laregely rewritten Problem formulation (Section 3) and Method (Section 4)
>   with clearer logic, wording and consistent notations
> - We have added Fig. 3 that shows a sample image of distracting domains.
> - We added Appendix
> 	- Pretrained policies' performance in source domain
>   - More detailed results on ablation study (scores by domain)
>   - Descriptions of network architectures
>   - Descriptions of hyperparameters
>
> > Minor: Typo: "training environment the Distracting"
>
> Thank you for spotting this! We have fixed this typo in the updated draft.
>
> > The meaning of the values in Table 1 are unclear. The legend has no information and the text says it's the reward difference between before and after adaptation, but then there's a column for before and another for after, so the numbers themselves are not the difference. What is it?
>
> Table 1 in original submission is moved to Table 2.
> The table compares the performances (episode returns) after adaptaion by PAD and ITI.
> This table shows that even though PAD helps, performances on ITI is consistently better across distraction types.
> We re-ran experiments to use the same pretrained policy in this experiment, and simplified the table.
>
> Please let us know if the updated paper and these answers address your concern.
>
> Also please feel free to ask us any questions!
>
> Thank you so much for thoroughly reviewing the paper and giving us thoughtful feedback, again!

---

### Official Review · Reviewer_jjRF · 2021-11-02

**Correctness:** 3
**Technical Novelty And Significance:** 3
**Empirical Novelty And Significance:** 3
**Recommendation:** 6
**Confidence:** 3

**Main Review:**

**Strengths:**

 * The approach itself is novel and relies on leveraging the model's ability to infer invariance via an adversarial loss on latent representations from source and target distributions. This provides a more principled way to model environment invariance and can possibly scale well in comparison to data driven approaches.

 * The approach does not rely on rewards from the target distribution.  However, it is unclear whether this is typically the case for the other baselines.

* The results showing decreased degradation for distraction on DMC (Fig. 5) look good and are consistent across baselines and distraction dimensions.  The results against PAD are also significantly in favour of ITI.

* A nice ablation demonstrating the effect of the adversarial objective.

**Weaknesses:**

* The evaluation domain seems to be narrow and focuses on robotics control domains or environments that simulate that.  How well does this approach work on other domains?  Are there any theoretical bounds or existing results that might give us confidence that this approach scales well across any out-of-distribution evaluation scenario?

 * The paper focuses on comparisons to data augmentation approaches to modeling out-of-distribution data.  Are there other approaches that might be considered?  Could combinations of this approach and data augmentation lead to even more robust adaptation to out-of-distribution estimation?

 * A more formal description of an adaptation and the criteria for being out-of-distribution would be very helpful.  In general, in sections 1 and 3 clearer explanation of the problem and definitions would lead to more clarity around the problem that this model proposes to solve.  For instance, what are the failure cases of generalization that this approach aims to solve and what in general are common features of a model with robust generalization?  What if we simply have access to large amounts of more varied data, do the gains of ITI remain strong in such scenarios?

 * First sentence of the last paragraph of intro, it is stated "this challenging scenario".  Can you be precise and explicit here?

 * The *Algorithm Overview* section seems like it could us more detail. An algorithm box could be useful here.  Do we use training samples only to train the encoder?

* Both the ablations and comparison to PAD are done over a fixed set of hyper-parameters.  It would be helpful if these conclusions were better supported across more variation and/or datasets.

* In the experiments section you emphasize that these methods "expand support for the training distribution", could you contrast these methods with each other?  Do they have access to reward of the target distribution?

* Table 1 doesn't seem to describe what is being compared.  Could you add more description here?

* When discussing the comparison to PAD, it 's stated that the hypothesis is that it does not encourage latent train and test distributions to match as ITI does, could this be described in more detail wrt to PAD itself?  Just a brief description would help lend insight as to why ITI performs better.



**Summary Of The Paper:**


**Introduction:**

In this paper the authors aim to tackle the problem of learning a model that generalizes well on a test distribution that samples outside of the training data distribution.  They aim to do this not via data augmentation but by using an inference model over the latent space of the input.  Such an approach needs to go beyond in-distribution generalization. If the target domain is not know a priori things like domain transfer, domain randomization and meta-learning techniques may fail.

Out-of-distribution generalization is very important but more challenging:
 * Prior experience will be critical
 * Lack a reward function over the test distribution
 * In robotics for instance, this type of learning is critical

The authors propose to: *"recast out-of-distribution generalization problem into an unsupervised policy adaptation between two MDPs that share a similar latent dynamics and reward structure, but with distinct observations"*

Statement of purpose for the paper: *"Harness probabilistic inference to produce invariance by taking advantage of latent structure"*  This work attempts to distinguish itself from other approaches that *"bake policy invariance"* in at training time.


**Problem Formulation:**

Given an MDP: M = <S, O, A, R, P, γ>, S contains ground truth info not always accessible to agent.  In particular two distinct MDPs: A train MDP M_train and a test MDP M_test.  The agent has no access to the reward function at test time.  The problem is defined: *"Often, we are most interested in the case where shift between M_train and M_test is induced by differences in the observation spaces O_train and O_test. That is, the state and reward structures between the train and test MDPs are quite similar, but the observation spaces between the two MDPs are significantly different."* It is noted that recent work in RL has tried to deal with this via data augmentation but this can lead to instability.

**Invariance Through Inference and Algorithm**

The goal here is to learn an encoder mapping semantically similar states in the train and test distribution to latent vectors.  This is to be accomplished given there is no access to rewards at test time and no access to paired trajectories (ie. matched test/train trajectories). The approach is defined by two objectives:
 1. a distribution matching objective encourages the latent distribution induced by both MDPs to match.
 2. a GAN-style loss to help ensure that the latent code is not using distracting information that would help distinguish the train and test MDPs.

The authors state that *"This adaptation objective is effectively maximizing the mutual information between the prototypical representation kept from during training, and the adapted representations of the new observations"* and further claim that *"our adaptation objective produce invariance by making statistical inference in the latent space."*  As regards the algorithm they also state that: *"The encoder G takes an observation from target environment, and learns to trick the discriminator, while the discriminator predicts whether the input comes from the encoder or source buffer."*

**Experiments**

The authors us the Deepmind Control Suite (DMC) as a training environment and the Distracting Control Suite (DistractingCS) for a test environment.  *"DistractingCS adds three types of distractions to the DeepMind Control Suite through deviations in background, camera pose, and color."*. Results are presented showing that baseline methods augmented with ITI show much slower performance degradation as the distraction intensity is increased.

The authors also compare ITI to Policy Adaptation during Deployment (PAD).  They compare to ITI on DMC for fixed distraction intensity.
ITI proves to yield stronger results in this case and it is posited that PAD does not encourage latent train and test distributions to match as ITI does: *We suspect this instability is caused by the large deviations in the latent variable distribution as a result of
changes in the target environment. In particular, we posit that the signal from PAD’s inverse dynamics head does not encourage the latent train and test distributions to match, which is a feature specifically baked into Invariance through Inference.*


**Summary Of The Review:**


While I believe that the ideas of this work are novel and very interesting and provide a potential pathway away from data augmentation method for out-of-distribution generalization I have two issues with the paper: 1) the evaluation domain may be too narrow to provide confirmation of the approach with confidence over other methods, 2) more clarity and details of the approach would be helpful in understanding this work, how it contrasts with others, and where it exceeds those.

Post Rebuttal:

My initial two main concerns with this work: 1) how well the approach generalizes and 2) the clarity of the paper. The updated draft addresses the second concern very well and so I consider that resolved. The sheer volume of updates here is impressive. For the first point, the authors have carried out a large number of new experiments which strengthens the case for control domains but to me the overall claims of the paper seems to be more general than that. That said, I believe this is a worthy contribution given the efforts put forth by the authors and will also increase my score to a 6.  Great work!

---

> ### Author Response · Authors · 2021-11-12
> **Thank you for your feedback & comments!**
>
> Thank you so much for your thoughtful feedback and comments!
>
> We agree that problem formulations and definitions can be written in a clearer way. We are in the process of updating the text to make the presentation clearer, and plan on uploading a revised version in a few days. We believe that updated problem formulation and more fine-grained experimental results can provide insights on when our approach works and when not.
>
> We'll also add algorithm box to clarify the exact procedures of the approach.
>
> In terms of the experimental results, our experiments are actually fairly comprehensive but we failed to present it in a clear manner. Figure 5 (d) for example, each point is actually aggregated from 5 domains, for 3 different modes of distractors, with 4 fairly complex baselines (3 shown in the submission): SVEA, DrQv2, SODA, and SAC, each with 5 random seeds. So each point is actually 300 runs!!
> But in addition to this, we plan to add another experiment that shows the sensitivity of our approach to hyperparameters more explicitly.
>
>
>
> **Other quick responses**
>
> **Do we use training samples only to train the encoder?**
> It depends on what you refer to by "training samples". If those refer to the observations of source environment (Figure 4), they are fed to the pre-trained encoder and we store the latents to source buffer.
> We note that the encoder here is pre-trained with a standard Deep RL method in the source environment (SAC, SVEA or DrQv2). We will clarify the description about this in the paper.
>
> **In the experiments section you emphasize that these methods "expand support for the training distribution", could you contrast these methods with each other? Do they have access to reward of the target distribution?**
> They differ in what types of augmentation to use (random crop, image overlay, random convolution, etc.) and where in training to use augmentation (only augment the Q-target, only augment the current Q function, etc.).
> They don't assume any specific target domain, and thus they have no access to reward of the target distribution. Essentially they simply train the agent with a broader range of observations, and that expands the support for the training distribution. As long as the test distribution falls within the expanded training distribution the methods should work. We will clarify this point in the paper.
>
> **Why is PAD (adaptation with inverse dynamics) not enough?**
> The inverse modeling objective forces the **local transition between two near-by states** to contain mutual information against the action. This is a local object that does not enforce **global consistency over the overall distribution**. We consider that the distribution matching in latent space via adversarial training encourages the global consistency (i.e., matching marginal distributions of z) and aligning both local and global structure is crucial to make the agent perform well in target environment. We are now re-writing the text to describe this clearer.
>
>
> Updating a paper is going to take us a few days, and we sincerely hope that you can take a look and update your evaluation after the update based on it. Thank you again for your thoughtful review!

---

> ### Author Response · Authors · 2021-11-18
> **Response (1/3)**
>
> (This response was edited for better clarifications)
>
> Dear reviewer jjRF,
>
> Thank you so much again for giving us thorough feedback and comments.
>
> In the following, we address each of your question / comment:
>
> > The evaluation domain seems to be narrow and focuses on robotics control domains or environments that simulate that. How well does this approach work on other domains?
>
> The authors agree with your concern, and have included an additional 4 control domains post-submission in addition to the 5 domains we had originally. This is about 160 new experiments that we have conducted in the past few weeks each taking rougnly 8 - 20 hours on a GPU-machine.
>
> With this additional 4 control domains, the amount of evaluation in this paper is now on-par with a number of recent publications. We offer comprehensive comparison against [^1, 2, 3, 4] as baselines, which is roughly 400 experiments in-total not counting sweeps that were needed for hyperparameter tuning. This is a very substantial amount of empirical evaluation.
>
> | Publication                 | Published Venue  | Domains                                      |
> | ----------------------------| ---------------- | -------------------------------------------- |
> | DrQ, Korstikov *et al* [^1] | ICLR 2020        | DeepMind Contron Suite, ALE (Not applicable) |
> | PAD  [^2]                   | ICLR 2020        | 9 DMC Domains, 1 Custom Domain (Not released) |
> | SODA [^3]                   | ICRA 2021        | 5 DMC Domains, 1 Custom Domain (Not released) |
> | SVEA [^4]                   | NeurIPS 2021     | 5 DMC Domains, 1 Custom Domain (Not released) |
> | ITI (Ours)                  |                  | 9 DMC Domains (5 at submisson, 9 during rebuttal) |
>
> [^1]: Kostrikov, I., Yarats, D. and Fergus, R. (2020) ‘Image Augmentation Is All You Need: Regularizing Deep Reinforcement Learning from Pixels’.
>
> [^2]: Hansen, N. et al. (2020) ‘Self-Supervised Policy Adaptation during Deployment’.
>
> [^3]: Hansen, N. and Wang, X. (2020) ‘Generalization in Reinforcement Learning by Soft Data Augmentation’.
>
> [^4]: Hansen, N., Su, H. and Wang, X. (2021) ‘Stabilizing Deep Q-Learning with ConvNets and Vision Transformers under Data Augmentation’.
>
> ---
>
> > Are there any theoretical bounds or existing results that might give us confidence that this approach scales well across any out-of-distribution evaluation scenario?
>
> The authors have considered the invariant risk minimization frame work [^5] and its extension to RL [^6] for a way to provide theoretical bounds. The main issue that we have ran into, is that these theoretically rigorous approaches make the assumption that the domain we deploy into needs to be known *a priori*. This is a weak assumption that we would like to remove in our Invariance Through Inference framework.
>
> That said, if we have a good way to define the ``intensity'' of variations, we could provide theoretical bounds. The challenge is that as demonstrated in Stone *et al*, even within the Distractor Control Suite, how difficult it is to generalize depends on the specific mode of distraction. Therefore the performance will depend on the deployment environment that we do not know *a priori*, and the architecture we use to parameterize the agent. Making providing a rigours bound a little difficult.
>
> The authors do consider this improvement on the theoretical front important.
>
> [^5]: Arjovsky, M. et al. (2019) ‘Invariant Risk Minimization’, arXiv [stat.ML]. Available at: http://arxiv.org/abs/1907.02893.
> [^6]: Sonar, A., Pacelli, V. and Majumdar, A. (2020) ‘Invariant Policy Optimization: Towards Stronger Generalization in Reinforcement Learning’.
>
> > The paper focuses on comparisons to data augmentation approaches to modeling out-of-distribution data. Are there other approaches that might be considered?
>
> Indeed! We consider data agumentation to be a way to produce perceptual invariance by exposing the agents to the variations during training. We refer to the over-all umbrella of methods as "invariance through interpolation". Meta-learning is an alternative approach that also falls under this category. A third approach is learning symmetries. We believe some of the data augmentation-based methods already does this, given the flat response that SVEA has on the "color" mode of distractors (Fig. 4 (c)). Namely the SVEA agent learned to ignore the color.

---

> ### Author Response · Authors · 2021-11-26
> **Response (2/3)**
>
> > Could combinations of this approach and data augmentation lead to even more robust adaptation to out-of-distribution estimation?
>
> Indeed! We spent a lot of effort experimenting with applying ITI to data-augmentation based methods such as SVEA and DrQ-v2 the result of which is included in the submission. This is exactly because we want to know if there is a synergy between using invariance through inference in-combination with policies that have been exposed to diverse data. The results on SVEA (Figure 4a) shows that data-augmentation still does a detrimental effect on policy performance (which is why the curve for SVEA starts at lower performance than DrQ-v2), but when combined, appying invariance through inference on an SVEA agent produces state of the art performance on these distractor control environments.
>
> > A more formal description of an adaptation and the criteria for being out-of-distribution would be very helpful. In general, in sections 1 and 3 clearer explanation of the problem and definitions would lead to more clarity around the problem that this model proposes to solve. For instance, what are the failure cases of generalization that this approach aims to solve and what in general are common features of a model with robust generalization? What if we simply have access to large amounts of more varied data, do the gains of ITI remain strong in such scenarios?
>
> We largely rewritten Problem formulation (Section 3) and Method (Section 4) to clarify the problem we are solving and its formulation.
>
> A typical failure case we often observe was that the distraction makes it hard for the encoder to estimate agent's state. For example, on "color" distraction, often agent's color becomes dark blue which is very similar to its background and floor. And on "backgroud" distraction, often distracted background has a yellow texture that is very similar to agent's original color. We believe that in these cases encoder fails to correctly predict agent's state, and thus ITI fails to adapt well.
> Successful scenarios include the domains that are easy to explore with random policy such as reacher-easy. As our method performs adaptation sampling from source and target buffers, it is crucial that these buffers have diverse set of observations. Thus, ITI works well on such domains whose state space is fairly limited and easy to explore.
>
> Again, we don't have a clear measure on "the amount of varied data", or "amount of augmentation", however, we believe that showing different augmentation-based approaches and their zero-shot performance can demonstrate the relationships between augmentations and ITI's gain.
>
> > First sentence of the last paragraph of intro, it is stated "this challenging scenario". Can you be precise and explicit here?
>
> We believe it is now clear that "this challenging scenario" refers to out-of-distribution generalization.
>
> > The Algorithm Overview section seems like it could us more detail. An algorithm box could be useful here. Do we use training samples only to train the encoder?
>
> We added algorithm box in Algorithm 1.
>
> We first pre-fill source buffer and target buffer, and then use samples from these to adapt encoder that is pretrained with a standard RL algorithm. The discriminator is jointly trained, but this one is from scratch.
> After the adaptation is done, we just plug in the adapted encoder to the agent, while reusing the pretrained policy.
>
> > Both the ablations and comparison to PAD are done over a fixed set of hyper-parameters. It would be helpful if these conclusions were better supported across more variation and/or datasets.
>
> We indeed ran every experiment with 5 random seeds (i.e., different initial weights and different distractions with the same intensity). But it was not very clear in the original text. We explicitly stated this in the new version.
> We also added more comprehensive ablations in Appendix C.

---

> ### Author Response · Authors · 2021-11-26
> **Response (3/3)**
>
> > In the experiments section you emphasize that these methods "expand support for the training distribution", could you contrast these methods with each other? Do they have access to reward of the target distribution?
>
> Any approaches on this paper does not have access to reward during adaptation on the target environment.
> DrQ-v2 and SVEA differ in what types of augmentation to use and how the augmented date is used for adaptation.
>
> DrQ-v2 is an extension of DrQ [^Kostrikov] which is one of the simplest methods in augmentations in deep RL, and this performs augmentation (random shift and crop) on both policy input and Q-function as well as Q-target.
> SVEA [^Hansen] shows that using augmented data in computing Q-target makes the optimization unstable. Thus it never uses augmentation for Q-target, and for policy and Q-function it mixes both augmented and unaugmented data to make optimization easier. It uses random shift and color jittering as its augmentations.
>
> [^Kostrikov] Kostrikov et al. (2021) ‘Image Augmentation Is All You Need: Regularizing Deep Reinforcement Learning from Pixels’, arXiv [cs.LG]. Available at: https://arxiv.org/abs/2004.13649
>
> [^Hansen] Hansen et al. (2021) 'Stabilizing Deep Q-Learning with ConvNets and Vision Transformers under Data Augmentation', arXiv [cs.LG]. Available at: https://arxiv.org/abs/2107.00644
>
>
> > Table 1 doesn't seem to describe what is being compared. Could you add more description here?
>
> Table 1 in original submission is moved to Table 2.
> The table compares the performances (episode returns) after adaptaion by PAD and ITI.
> This table shows that even though PAD helps, performances on ITI is consistently better across distraction types.
> We re-ran experiments to use the same pretrained policy in this experiment, and simplified the table.
>
>
> > When discussing the comparison to PAD, it 's stated that the hypothesis is that it does not encourage latent train and test distributions to match as ITI does, could this be described in more detail wrt to PAD itself? Just a brief description would help lend insight as to why ITI performs better.
>
> PAD pretrains the SAC agent with inverse dynamics auxiliary task, and during adaptation, it updates weights solely by gradients from inverse dynamics head.
> As we discuss in updated ablation study (Section 5.3), we theorize that inverse dynamics, or dynamics consistency loss, can only help adjusting the local pairwise consistency in the source and target latent spaces given that two spaces are already close with each other.
> But when the latent spaces differ largely, which is often the case when observations are largely shifted, PAD does not have an objective that explicitly matches the global structures of the latent spaces (marginal distributions of latent z). We believe lacking this component makes it challenging to align latent spaces.
>
> Please let us know if the updated paper and these answers address your concern.
>
> Also please feel free to ask us any questions!
>
> Thank you so much for thoroughly reviewing the paper and giving us a thoughtful feedback, again!

---

### Official Review · Reviewer_wVDQ · 2021-11-03

**Correctness:** 3
**Technical Novelty And Significance:** 3
**Empirical Novelty And Significance:** Not applicable
**Recommendation:** 6
**Confidence:** 2

**Main Review:**

## Strengths

* **Novelty**. As far as I am aware the authors propose a novel method for solving the problem of generalisation to new (but similar) domains in RL.

* **Relevance**. This is an important problem to solve for the real world deployment of reinforcement learning agents.

* **Empirical results**. The results provided indicate that the authors' method performs well.


## Weaknesses

* **Clarity**. Throughout the paper there were notation and wording choices which hinder clarity, in my opinion. I list them here:
    1. The phrase "ex-ante" is unnecessarily fancy, for lack of a better word. A more commonly used term would be better.
    2. The statement "... with different forward dynamics, so long as the distinct dynamics maintain *some semblance of similarity*" is very vague. Could it not be replaced with something more concrete?
    3. Regarding notation in Sec 4.1.
         A. In some cases G_\theta has two arguments, the observation and action, and in other cases it only has the observation. Which is correct?
         B. (Related to A), should the text after eqn 2, also mention a mapping similar observations *and actions* to close latents?
         C. Perhaps it would be helpful to write \hat{P}_\phi rather than P_\phi in eqn 2, to help distinguish P_\phi from the P in the MDP?
    4. \tilde{z} and D are not introduced before eqn 3.
    5. D_{dist} and D_{non-dist} in eqn 4 are never explicitly described.
    6. In eqn 5, it is not clear why J_inv is within the arg max over D, given that it does not depend on D (i.e. D does not appear in eqn 2)?
    7. Section 4.2 would be much more clear if written in an algorithm environment rather than as paragraphs of text.
    8. In figure 4, should 2. not contain P_\phi rather than D?


* **Experimental evaluation.**
    1. It seems to me that the experimental evaluation in this paper is not very comprehensive, only showing two main experiments and a single ablation. I would expect to see a wider range of experiments in an ICLR paper introducing a new method. Otherwise, how can readers know whether the method is generally applicable?
    2. Regarding the modifications to DistractingCS, it would be great to see the effects of the various combinations of 2 distractions.
    3. Why was the comparison to PAD not also made in Figure 5? It would be informative to compare how the degree of distraction impacts PAD.
    4. How many repetitions were made to get the confidence intervals in Table 1?
    5. The proposed future work of exploring the effect of exploration strategy seems like it should be included in this paper, in order to paint a full picture of the proposed method.

## Other comments/questions for the authors

1. A relevant but missing citation/discussion in the related work (specifically in **Invariant representation learning**) seems to be World Models (Ha and Schmidhuber, 2018).

2. (nit) Regarding the slight abuse notation for eqn 1, there seems to be enough space to write things correctly. Furthermore, the explanation for the abuse of notation takes more extra space than the correct notation would.

3. Regarding the discussion of data augmentation techniques in the paragraph above Sec 4. While it is true that even extreme data augmentation techniques would fail to cover the test MDP observation space, it is not clear that data augmentation does not ultimately play the same role as *invariance through inference* albeit in a less direct manner. That is, data augmentation should force the policy learned to be invariant to the noise from the data augmentations and thus be more likely to generalise to the test MDP setting.

4. Could the authors elaborate on their choice of the GAN framework as opposed to VAEs or some other deep generative model?

5. How much of a challenge was it to get the adversarial training to converge in the various experiments presented in the text. GANs are well known to be difficult to train and often require many tricks to converge reliably. This was not touched on in the manuscript and seems an important detail for the general applicability of this work.


## References

David Ha, Jürgen Schmidhuber:
World Models. CoRR abs/1803.10122 (2018)


**Summary Of The Paper:**

This paper proposes to learn the invariances relating to MDPs with different observations (e.g. invariances in the underlying dynamics of the systems) using a generative model of the observations in an unsupervised manner. The inference in this generative model is accomplished using a GAN framework with auxiliary losses for the inverse dynamics of the system.

**Summary Of The Review:**

While the authors' proposed method seems like a promising solution to a challenging and important problem, this paper is let down by a lack of clarity in presentation, and a narrow range of experimental evaluation.

++++ Post-revision update ++++

The authors have largely rewritten the paper which has resulted in a much clearer presentation. They have also expanded on their experimental results and clarified that the scope of their experimental work is larger than I initially understood. With these two points in mind, I have increased my score from 5 to 6. Unfortunately, I do still feel that the experimental evaluation could be more comprehensive, which is why I have not increased my score further. Additionally, I have not had the time I would like to go through the heavily revised paper, thus I have decreased my confidence.

---

> ### Author Response · Authors · 2021-11-12
> **Thank you for your feedback & comments!**
>
> Thank you for your thoughtful review! We would like to apologize for the lack of clarity in our writing. Since submission, we have re-written the method section, and are in the process of updating the way we present the experimental results. We have also uploaded our experimental code base as part of the supplementary material, which we plan to release on GitHub as soon as our paper is accepted.
>
> In terms of the experimental results, our experiment is actually fairly comprehensive but we failed to present it in a clear manner. Figure 5 (d) for example, is actually aggregated from 5 domains, for 3 different modes of distractors, with 4 fairly complex baselines (3 shown in the submission): SVEA, DrQv2, SODA, and SAC, each with 5 random seeds. So this is actually 300 runs!!
>
> We are sorry for the lack of clarity in our presentation of the results. To fix this issue, besides a few additional ablation experiments we are currently running, we are breaking down the results into smaller pieces, and showing slices through select domains and modes of distractions, to make the comparison much clearer.
>
> This update is going to take us a few days to upload here, but we sincerely hope that you could take a look and update your evaluation. Thanks again for your thoughtful review! We will also address some of the specific points while we are updating the draft.

---

> ### Author Response · Authors · 2021-11-18
> **We have significantly updated section 1 - 4 on the paper.**
>
> Dear reviewer wVDQ,
>
> Thank you so much again for giving us thorough feedback and comments.
>
> Following your feedback, we have significantly updated sections 1 - 4 on the paper,
>  restructuring the text to clarify the ideas and narratives.
> We have also fixed wrong or inconsistent notations that you have kindly pointed out altogether.
>
> We believe this update addresses many of the concerns and flaws you mentioned.
> Especially we stopped using unnecessarily fancy phrasing and tried to be more straight-to-the-point.
>
> We put so much time into updating the paper following your kind feedback.
> So we would sincerely appreciate it if you could spend some time going through the updated sections 1 - 4,
> and edit the score accordingly if you believe the paper got improved by addressing your concerns.
>
> NOTE: we have **not** edited the Experiment section at this point yet.
> But we are working very hard to update some of the experiment results right now. We should be able to reflect those results in a few days.

---

> > ### Comment · Reviewer_wVDQ · 2021-11-20
> > **A little help**
> >
> > Thanks very much for the updates so far. It would be very helpful to me if you could respond to each of my points directly to explain the changes you made and where I can find them in the updated manuscript. This will make it much easier for me to evaluate whether or not my concerns have been addressed and thus whether or not to increase my score. I am looking forward to seeing the new experimental section too.

---

> > > ### Author Response · Authors · 2021-11-20
> > > **We clarify what changes are made for your questions and comments**
> > >
> > > Dear reviewer wVDQ,
> > >
> > > Thank you for your response!
> > > We address / clarify what changes are made for your questions / comments:
> > >
> > > ### Clarity
> > >
> > > > The phrase "ex-ante" is unnecessarily fancy, for lack of a better word. A more commonly used term would be better.
> > > > The statement "... with different forward dynamics, so long as the distinct dynamics maintain *some semblance of similarity*" is very vague. Could it not be replaced with something more concrete?
> > >
> > > We replaced the sentence that included "ex-ante" (Paragraph 3 in Section 1).
> > > We also replaced the sentence that included the vague explanation with "semblance of similarity" (The last paragraph in Section 1)
> > >
> > > > Regarding notation in Sec 4.1. A. In some cases G_\theta has two arguments, the observation and action, and in other cases it only has the observation. Which is correct? B. (Related to A), should the text after eqn 2, also mention a mapping similar observations *and actions* to close latents? C. Perhaps it would be helpful to write \hat{P}_\phi rather than P_\phi in eqn 2, to help distinguish P_\phi from the P in the MDP?
> > > > \tilde{z} and D are not introduced before eqn 3.
> > > > D_{dist} and D_{non-dist} in eqn 4 are never explicitly described.
> > >
> > > We restructured the problem formulation and method section.
> > > $G_\theta$ (encoder) is replaced with $F$, and the notation is used consistently.
> > > Trajectories are sampled from $P_{\bar{\pi}}(\mathcal{M})$ rather than $D_{dist}$ and $D_{non-dist}$ and we clearly stated the definitions in the text.
> > >
> > > > In eqn 5, it is not clear why J_inv is within the arg max over D, given that it does not depend on D (i.e. D does not appear in eqn 2)?
> > >
> > > We still have the same equation in Eq. (7). You understand it absolutely correctly in that J_dyn does not depend on discriminator D, and thus J_dyn does not have to be in argmax_{D}. However, we cannot simply pull D out of the bracket as the equation will be
> > >
> > > argmax_F [ argmax_D [ J_adv ] + J_dyn ],
> > >
> > > which is totally incorrect. For the simplicity and correctness, we keep the form of Eq. (7).
> > >
> > > >  Section 4.2 would be much more clear if written in an algorithm environment rather than as paragraphs of text.
> > >
> > > We fully agree. We added an algorithm box on page 6.
> > >
> > > >  In figure 4, should 2. not contain P_\phi rather than D?
> > >
> > > Exactly. This was our mistake. We end up removing figure 4 as this step is now clear from the text and newly added algorithm box.
> > >
> > > ### Other comments/questions for the authors
> > >
> > > >  A relevant but missing citation/discussion in the related work (specifically in **Invariant representation learning**) seems to be World Models (Ha and Schmidhuber, 2018).
> > >
> > > We believe that their focus was on learning a good latent representations that can simulate the world, rather than learning representations that stays invariant across domains.
> > >
> > > > (nit) Regarding the slight abuse notation for eqn 1, there seems to be enough space to write things correctly. Furthermore, the explanation for the abuse of notation takes more extra space than the correct notation would.
> > >
> > > We totally agree. We removed Eqn (1) as it was a bit too general to explain our approach. We made sure that notations are defined and consistent throughout the paper.
> > >
> > > > Regarding the discussion of data augmentation techniques in the paragraph above Sec 4. ... That is, data augmentation should force the policy learned to be invariant to the noise from the data augmentations and thus be more likely to generalise to the test MDP setting.
> > >
> > > We consider that data-augmentation scheme only works when the observations in the target (test) domain falls within the augmentation distributions. Thus, given an augmentation scheme, we can always add an *adversarial* distraction to the domain that makes it out-of-distribution. As another aspect, including larger amount of augmentation makes policy training harder. Thus we believe unsupervised adaptation such as our approach truly makes out-of-distribution possible.
> > >
> > > >  Could the authors elaborate on their choice of the GAN framework as opposed to VAEs or some other deep generative model?
> > >
> > > That is actually a great point. We haven't tried other methods to match latent distributions. We chose adversarial training as it is a commonly-used approach. One possible drawback in using VAE could be that it requires to define a specific form of a prior, which may enforce undesirable constraint.
> > >
> > > >  How much of a challenge was it to get the adversarial training to converge in the various experiments presented in the text. GANs are well known to be difficult to train and often require many tricks to converge reliably. This was not touched on in the manuscript and seems an important detail for the general applicability of this work.
> > >
> > > We use Wasserstein-GAN like architecture to encourage its stability in training. We didn't see catastrophic collapse or divergence during training.
> > >
> > > For the questions / comments on experiments section, we will respond to each when we update the paper (we will shortly).

---

### Author Response · Authors · 2021-11-23
**Thank you very much for your useful feedback.**

Dear reviewers,

As the discussion period ends today, we again sincerely want to thank all of you, reviewers, for carefully reading through our draft and providing insightful feedback.
We really appreciate that you spent your precious time on this, and thanks to your comments, we could significantly improve our draft.

Ever since the reviews came out, we have worked really hard on improving the draft by a significant margin.
At this point, we are quite confident that the current draft would qualify for top ML venues.
To summarize the changes we made:

### Section 3: Problem formulation
- We have largely re-written the text with clearer and consistent notation
- We replaced the unnecessarily general formulations with rather concrete ones to avoid confusion

### Section 4: Invariance Through Inference
- We have largely re-structured the section by introducing two separate subsections explaining each objective
- We also added the algorithm box (Algorithm 1) to further clarify the procedures of our approach

### Section 5: Experiment
- We added a sample image of our distraction suite (Figure 3)
- We added a table that shows domain-wise adaptation performance (Table 1)
- We added learning curves during adaptation (Figure 5)
- We added more discussions regarding ablation results

### Appendix
- We added a domain-wise performance of baseline policies in non-distracting environment
- We added the more detailed table for ablation studies
- We added descriptions of network architectures and hyperparameters

We hope very very strongly that you can at least go through the updated draft. Again, we have spent so much time on enhancing the paper, by following reviewers' comments. And we are confident that the draft has gotten significantly better.
We would be terrifically grateful if you could update the score so that it aligns with the **current draft** but not the original draft.

---

### Decision · Program_Chairs · 2022-01-20

**Decision:**

Reject

**Comment:**

Meta Review for Invariance Through Inference

The motivation of this work is to address the problem of learning a model that generalizes well on a test distribution that samples outside of the training data distribution. Reviewer X3P2 wrote a good summary of the paper:

In this paper, the transfer of a reinforcement model (RL) setting from an idealized (training) environment to a more realistic environment with distractors in the observations is considered. Instead of augmenting the training environment with more data so as to make the system more resilient to variations and distractors, the system is adapted at test time to be invariant to the specific distractors found in the environment. Experiments in simulation show the benefits of the proposed approach. Crucially, the agent is not able to access any reward data at test time.

Reviewers, including myself, recognize the novelty of the approach, in particular appreciate the authors' motivation to provide a more principled way to model environment invariance that can possibly scale well in comparison to data driven approaches. However, the initial round of feedback is generally negative, in particular, most reviewers raise concerns regarding the lack of clarity in presentation, and also have issues with the narrow range of experimental evaluation. Clearly, this is promising work, but possibly had to be rushed for submission.

To the authors' credit, they devoted substantial efforts to completely revamp their paper, addressing many of the issues head on. The resulting updated manuscript is almost a complete rewrite of the paper. All reviewers acknowledge (and praise) the effort from the authors' to improve the paper, and 3 out of 4 reviewers had improved (or maintained) their scores from rejection to a 6. But as the paper is a complete revamp, reviewers did not have the time to assess the entire rewrite of the paper (it's like the need to review a paper from scratch), so the confidence is reduced.

While X3P2 did not change their score, they did lead a discussion amongst myself and other reviewers, and they spent the time to take a detailed look at the completely revised draft. Here are the comments from that discussion, for full transparency:

---

*The paper has been completely revamped, to the point in which the presented technique is actually different (the dynamics loss now includes a new forward term). The changes are overall welcome since it significantly improves the clarity of the presentation. Experiments still show promise.*

*I still have problems with the theoretical aspect of it, though. I think that it is unclear why the proposed system is working and fails to provide the minimal system that works.*

*Equation (4) is dimensionally incorrect. It's summing squared error over actions with squared error over latents. Both of these are arbitrary units that can lead to the forward or backward losses dominating. It's also unclear why both of these losses are necessary and not just the inverse one.*

*Equation (7) is similarly dimensionally incorrect. Again, units are arbitrary and for all we know the joint loss could be ignoring the dynamics loss or the adversarial loss.*

*The use of a GAN and the corresponding loss is unjustified. As the authors acknowledge, if the dynamic loss is very small, then the system should already work. They argue that finding that parameterization without the adversarial loss is challenging. There's a difference between using the right loss and finding the right way to optimize it, but here it seems that the right loss is being modified for optimization purposes. Is the adversarial loss something we really want to minimize or just something that helps find the best dynamics loss? What if we remove the adversarial loss after being close to convergence? What if we use multiple restarts or other techniques to help with the optimization of only the dynamics loss? My take from the theory is that the adversarial loss term shouldn't be needed and that the challenging optimization problem should be addressed (rather than modifying the loss).*

*The new ablation experiments are also confusing: If the dynamics loss is the actual driver, and the adversarial loss only helps with finding a good solution, how come that we get almost equally good results when we remove the dynamic loss? Matching the latent distribution shouldn't be enough to have aligned latents. Maybe there's something about the architecture of F that matches the ground truth, so that matching the distribution aligns the latents. This hints towards the adversarial loss actually playing an important role beyond helping with the optimization problem. This is not supported at all by the theory, since matching distributions should result in arbitrary latents and potentially performance of a random system. In fact, given the problems with units, the joint loss might be dominated by the adversarial loss, which would explain this result.*

*It seems like there's something here, but I think more work is necessary to really understand which pieces are necessary in this system and whether there's some sort of adaptation between the experimental setup and F that would explain why distribution matching results in latent alignment, which is not expected (Zhu et al. 2017). Also, the units problem makes the ablation results even harder to interpret: Maybe the dynamics loss is playing a small role in the joint loss, and that's why removing it completely has a small impact.*

---

After much assessment, while I do find this work to be interesting and potentially highly impactful (since they introduce an alternative approach to data-centric one OOD), the final manuscript's assessment is still borderline (the reviewers all mentioned that while they recognize the improvement, they list issues from preventing their full endorsement), and X3P2 still found several issues with the revision (which I do believe can be addressed in due time). While I'm fully confident that with additional work, this paper could have the potential to be an impactful one, I am currently on the side of not recommending it for acceptance for ICLR 2022.

Note to the PC's, that this is a borderline decision. If the PC's want to flip the decision to an accept, and think the post rebuttal issues are small enough, I'll be fine with that. But in any case, I look forward to seeing a further improved version (of the revamped manuscript) published in a journal or presented at a future conference. Good luck!